# A fungal metabolic regulator underlies infectious synergism during *Candida albicans-Staphylococcus aureus* intra-abdominal co-infection

Saikat Paul[1,11], Olivia A. Todd [2,11], Kara R. Eichelberger[3], Christine Tkaczyk [4], Bret R. Sellman[4], Mairi C. Noverr[5], James E. Cassat[3,6,7,8], Paul L. Fidel Jr[9] & Brian M. Peters [1,10] ✉

*Candida albicans* and *Staphylococcus aureus* are two commonly associated pathogens that cause nosocomial infections with high morbidity and mortality. Our prior and current work using a murine model of polymicrobial intra-abdominal infection (IAI) demonstrates that synergistic lethality is driven by *Candida*-induced upregulation of functional *S. aureus* α-toxin leading to polymicrobial sepsis and organ damage. In order to determine the candidal effector(s) mediating enhanced virulence, an unbiased screen of *C. albicans* transcription factor mutants was undertaken revealing that *zcf13Δ/Δ* fails to drive augmented α-toxin or lethal synergism during co-infection. A combination of transcriptional and phenotypic profiling approaches shows that *ZCF13* regulates genes involved in pentose metabolism, including *RBK1* and *HGT7* that contribute to fungal ribose catabolism and uptake, respectively. Subsequent experiments reveal that ribose inhibits the staphylococcal *agr* quorum sensing system and concomitantly represses toxicity. Unlike wild-type *C. albicans*, *zcf13Δ/Δ* did not effectively utilize ribose during co-culture or co-infection leading to exogenous ribose accumulation and *agr* repression. Forced expression of *RBK1* and *HGT7* in the *zcf13Δ/Δ* mutant fully restores pathogenicity during co-infection. Collectively, our results detail the interwoven complexities of cross-kingdom interactions and highlight how intermicrobial metabolism impacts polymicrobial disease pathogenesis with devastating consequences for the host.

*Candida albicans*, an opportunistic fungus, and *Staphylococcus aureus*, a ubiquitous bacterial pathogen, are among the top causes of serious nosocomial infections and invasive diseases[1,2]. While these microbes can cause significant morbidity and mortality on their own, they are often co-isolated from various colonization and infection niches and are correlated with more severe disease states and higher mortality rates, even with therapeutic intervention[3–8]. Cross-kingdom interactions between these microbes are shaped by physical binding, chemical signaling, and environmental factors, which may alter their pathogenicity[9–11]. Metabolic adaptation to the dynamic polymicrobial microenvironment also plays a key role. For example, it was previously shown that *C. albicans* amino acid catabolism can

indirectly elevate *S. aureus* toxin production and that bacterial peptidoglycan recycling and subsequent N-acetylglucosamine release is a feasible mechanism to stimulate *C. albicans* hyphal growth[10,12].

Polymicrobial infections of the abdominal cavity are a commonly investigated paradigm for microbe-microbe interactions that regulate disease outcomes. Intra-abdominal infections (IAI) are a collection of diseases characterized by microbial invasion and inflammation of the abdominal space. The introduction of microbes into the abdominal cavity generally results from trauma, such as perforations to the gastrointestinal tract, invasive surgery, and contamination of indwelling catheters. IAI can lead to more complicated infections, like sepsis, and are the second-most common cause of infectious mortality in ICU patients[13–15]. Fungal-bacterial polymicrobial IAIs result in more severe disease and increased mortality (up to 80%) as compared to 10–30% mortality during mono- or poly-bacterial IAI[3,16–19]. *C. albicans* and *S. aureus* rank among the most common etiological agents of IAI, including those associated with peritoneal dialysis (PD) catheter use[20]. Polymicrobial PD-related infections are associated with higher recurrence rate, catheter loss, and permanent switch to hemodialysis as compared to monomicrobial infection[21–25]. Additionally, *C. albicans* has been identified as an independent risk factor for mortality during IAI[17].

In support of these clinical data, a mouse model of polymicrobial IAI using *C. albicans* and *S. aureus* revealed a striking lethal synergism, where co-inoculation rapidly resulted in nearly 100% mortality, while monomicrobial infections were non-lethal[26]. *S. aureus* produces a number of toxins, primarily regulated through the *agr* quorum sensing locus[27]. Previous work from our laboratory has demonstrated that *C. albicans* can augment *S. aureus agr* activity and upregulate Agr-regulated genes during polymicrobial growth in vitro and during murine IAI, including *hla* that encodes for α-toxin. Moreover, α-toxin was required for lethality during polymicrobial IAI[28]. This virulence determinant oligomerizes into a heptameric β-barrel and is capable of non-specifically forming pores in many cell types, including erythrocytes, epithelial cells, endothelial cells, and various immune cells. In addition, α-toxin mediates platelet aggregation, leading to excessive microvascular clotting and thrombocytopenia observed in staphylococcal sepsis that contributes to pathological liver and kidney damage[29,30].

While it has been established that *C. albicans* is able to enhance staphylococcal α-toxin production in vitro and in vivo, the mechanism(s) by which *C. albicans* achieves this are incompletely defined[28]. We have previously determined that a candidal secreted factor is not likely directly stimulating the AgrC receptor of *S. aureus* to enhance α-toxin production[12]. While *C. albicans* Stp2p-mediated extracellular alkalinization contributed to in vitro *agr* activation, a *stp2Δ/Δ* mutant exhibited no pathogenicity defects during murine polymicrobial IAI[12]. Therefore, additional genetically encoded or physiological stimuli must be required for synergistic lethality.

The objective of this study was to further clarify the mechanism driving synergistic lethality during *C. albicans-S. aureus* polymicrobial IAI by identifying novel regulators of *Candida*-induced *agr* activation. An unbiased *agr* reporter co-culture screen revealed the uncharacterized candidal transcriptional regulator Zcf13p as a key factor required for staphylococcal *agr* activation and synergistic lethality during IAI. Further, we show that Zcf13p contributes to pentose sugar metabolism by controlling the expression of ribokinase (*RBK1*) and a homologous low-affinity ribose transporter (*HGT7*). Pentose sugars, including ribose, were found to inhibit the *agr* quorum sensing system and toxin production. Therefore, we present a mechanism by which *C. albicans* ribose metabolism derepresses *agr* signaling to drive *S. aureus* toxicity and lethality during IAI.

## Results

### α-toxin is responsible for exacerbated organ damage during polymicrobial IAI

Based on prior reports of organ damage caused by α-toxin in a model of staphylococcal sepsis, we wished to determine whether α-toxin similarly contributed to organ dysfunction during polymicrobial IAI[29,30]. Mice were challenged intraperitoneally with *C. albicans* and *S. aureus* wild-type (WT) or an α-toxin-deficient mutant (*hla::bursa*) and biomarkers of organ damage were kinetically assessed in the serum. At 12 h, polymicrobial infection with WT *S. aureus* led to significant increases in 3 common liver enzymes (alkaline phosphatase, ALP (Fig. 1a); alanine transaminase, ALT (Fig. 1b); aspartate aminotransferase, AST (Fig. 1c)) and in the kidney biomarker blood urea nitrogen (BUN) (Fig. 1d) as compared to co-infection with the *hla::bursa* strain. These results demonstrate that α-toxin is required for driving increased organ damage during polymicrobial IAI.

### Functional α-toxin is required for lethal synergism

α-toxin is secreted as monomeric units but adopts a heptameric pore in the cell membrane, which ultimately leads to its lytic activity[31–33]. Prior studies have demonstrated that the H35 residue in the N-terminus is essential for the stabilization of the heptamer and amino acid changes at this residue abolish lytic activity[34,35]. Although mutated α-toxin cannot form pores, it is still able to bind to membrane-bound ADAM10 and could potentially activate numerous intracellular signaling cascades to drive pathogenicity. To determine whether pore-forming activity was required for lethality during IAI, we constructed a plasmid containing the *hla* locus (pSK-*hla*) and used site-directed mutagenesis to introduce a single nucleotide change (182 A > T) leading to a nonsynonymous substitution in amino acid sequence (H35L). Plasmids containing the native or mutated *hla* loci were transformed into the *hla::bursa* mutant. As expected, supernatant from *hla::bursa*-p*hla* led to robust hemolytic activity using a blood agar lysis assay, while that from *hla::bursa*-p*hla*[H35L] led to no observable toxicity (Fig. 2a) despite similar α-toxin production from both strains (Fig. 2b). Both α-toxin isoform levels were increased when co-cultured with *C. albicans* (Fig. 2b) without overt growth differences between these strains during mono or polymicrobial culture (Fig. 2c).

We next sought to determine whether *hla::bursa*-p*hla*[H35L] lost the capacity to drive synergistic lethality during polymicrobial IAI with *C. albicans*. Mice were infected intraperitoneally (IP) with WT *C. albicans* and WT, *hla::bursa*, *hla::bursa*-p*hla*, or *hla::bursa*-p*hla*[H35L] *S. aureus* and survival was followed for up to 5 days post-infection (p.i). Mice co-infected with WT *S. aureus* or *hla::bursa*-p*hla* succumbed within 24 h, whereas co-infections with *hla::bursa* or *hla::bursa*-p*hla*[H35L] were non-lethal (Fig. 2d). Notably, there were no significant differences in the microbial burden (Fig. 2e) or amount of α-toxin found (Fig. 2f) in the kidneys, spleen, and peritoneal lavage fluid of mice co-infected with the two complemented strains at an 8 h endpoint. These data indicate that the oligomerization and cytolytic activity of α-toxin is required to drive lethality during polymicrobial IAI.

### The *C. albicans* transcription factor Zcf13p is required for enhancing *S. aureus* α-toxin production and driving lethal synergism

While staphylococcal α-toxin is necessary for lethality, the mechanism(s) by which *C. albicans* augments its production both in vitro and in vivo remained incompletely defined. Despite the impact *C. albicans*-mediated alkalinization has on *S. aureus agr* activity in vitro, an alkalinization-deficient mutant was still able to induce lethal synergism during polymicrobial IAI[12]. Therefore, to identify other potential candidal factors involved in enhancing α-toxin production, an unbiased screen was undertaken. Mutants from a *C. albicans* non-essential transcription factor (TF) deletion library were co-cultured with the *S. aureus agr*-GFP reporter strain *S. aureus*(pDB22) as described

previously[12,28,36,37]. Fold-change fluorescence of polymicrobial cultures was normalized to *S. aureus*(pDB22) mono-culture and z-score calculated (Fig. 3a, Supplementary Data 1). The TF library control strain (indicated by green dots, henceforth "TF WT") displayed a consistent 2−2.5-fold increase in GFP signal relative to mono-culture. We found 9 mutants that failed to enhance *agr* activity (≥2 standard deviations of the mean, gray dotted lines) to the same extent as the TF WT strain. Follow-up assays confirmed that these mutants exhibited a defect in enhancing *agr* activity (Fig. 3b) and α-toxin production (Fig. 3c). CFU counts from polymicrobial cultures revealed that *bas1Δ/Δ*, *leu3Δ/Δ*, and *msn4Δ/Δ* had significant growth defects that likely explained their inability to enhance *agr* activity (Fig. 3d); therefore, these mutants were excluded from further analysis. As the *agr* quorum sensing system is responsive to pH, we determined whether the identified mutants had alkalinization defects that could explain their inability to enhance *agr* activity[12,38,39]. Mutants *sfl1Δ/Δ*, *grf10Δ/Δ*, and *isw2Δ/Δ* exhibited significant alkalinization defects that likely led to attenuated *agr* activation (Fig. 3e).

We next evaluated whether the transcription factor mutants without growth defects could drive synergistic lethality during polymicrobial IAI. Mice were infected with WT *S. aureus* and either the TF WT or deletion mutants and followed for survival. Although most of the mutants elicited lethality like WT, *sfl1Δ/Δ* (50% mortality) and *zcf13Δ/Δ* (0% mortality) exhibited attenuated virulence during co-infection (Fig. 4a). We evaluated the microbial burden in the kidneys, spleen, and peritoneal lavage fluid of mice co-infected with TF WT, *sfl1Δ/Δ*, or *zcf13Δ/Δ* strains at 8 h p.i. and found no significant

differences (Fig. 4b). Despite similar growth, the *zcf13Δ/Δ* mutant was unable to enhance *S. aureus* α-toxin production in the kidneys and peritoneal cavity to the same level as the TF library control strain (Fig. 4c). This data demonstrates that *C. albicans* Zcf13p underpins modulation of *S. aureus* virulence in vitro and during polymicrobial IAI.

### *ZCF13* is necessary for *Candida*-induced *agr* activation

To confirm results obtained with the library deletion strain, independent mutant (SC *zcf13Δ/Δ*) and revertant (SC *zcf13Δ/Δ+ZCF13*) strains were constructed in the SC5314 background using previously published methods[40,41]. We confirmed that the newly constructed *zcf13Δ/Δ* mutant was deficient in enhancing *agr* activity and α-toxin during coculture and that reversion of *ZCF13* restored phenotypes to WT levels (Fig. 5a, b). No major growth defects between these strains were observed during mono- or co-culture (Fig. 5c). Additional phenotypic profiling for hyphal growth defects or stress susceptibility revealed no major differences between *zcf13Δ/Δ* and WT as similarly reported[36] (Supplementary Figs. 1 and 2). Co-infection with *S. aureus* and *zcf13Δ/Δ* did not display early infectious synergism and mice survived significantly longer as compared to co-infection with SC5314 or *zcf13Δ/Δ+ZCF13* (Fig. 5d). As anticipated, *zcf13Δ/Δ* was unable to enhance α-toxin production to the same levels observed with SC5314 or *zcf13Δ/Δ+ZCF13* (Fig. 5e). Importantly, microbial burdens in the kidneys, spleen, or peritoneal lavage fluid were similar between the three strains (Fig. 5f). These data indicate a crucial role for Zcf13p in driving early infectious synergism observed during polymicrobial IAI caused by *C. albicans* and *S. aureus*.

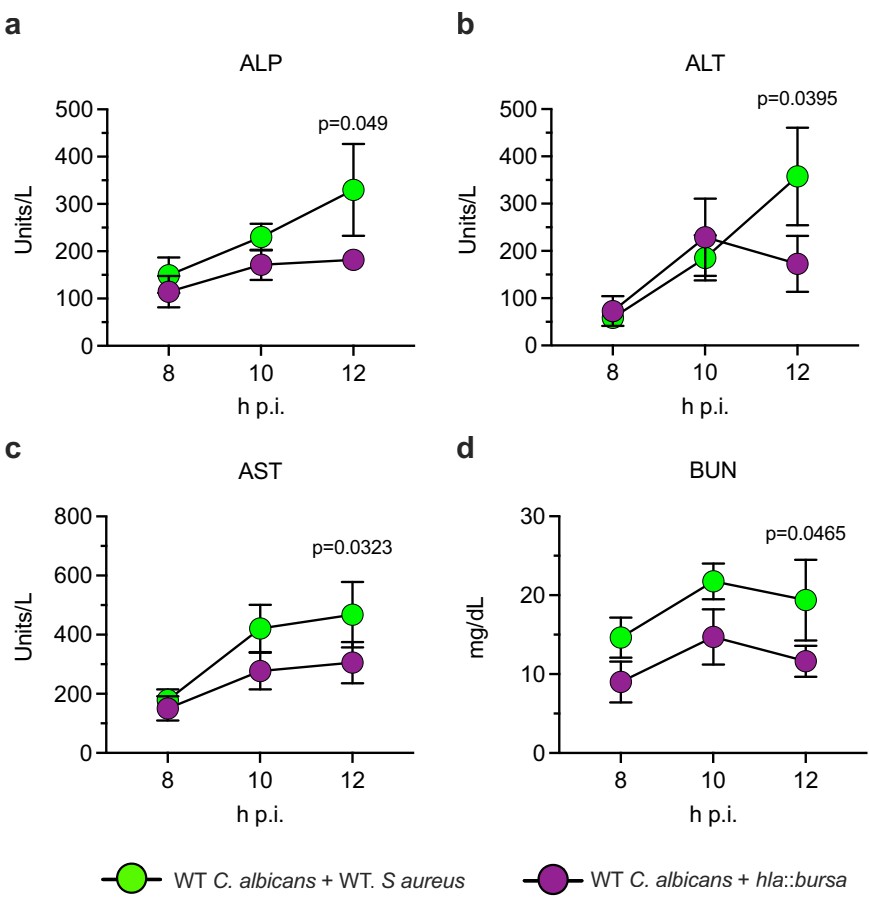

**Fig. 1 | Staphylococcal α-toxin drives organ damage during IAI.** Mice (*n* = 8 per group) were infected with WT *C. albicans* + WT *S. aureus* (green) or *hla::bursa* (purple) and sacrificed at 8, 10, or 12 h post-infection. Levels of serum **a** alkaline phosphatase (ALP), **b** alanine transaminase (ALT), **c** aspartate aminotransferase (AST), and **d** blood urea nitrogen (BUN) were assessed. Data are depicted as the mean ± SEM. Significance was determined by comparing WT *C. albicans* + WT *S. aureus* with WT *C. albicans* + *hla:bursa* at each time point using a two-sided unpaired multiple t-test.

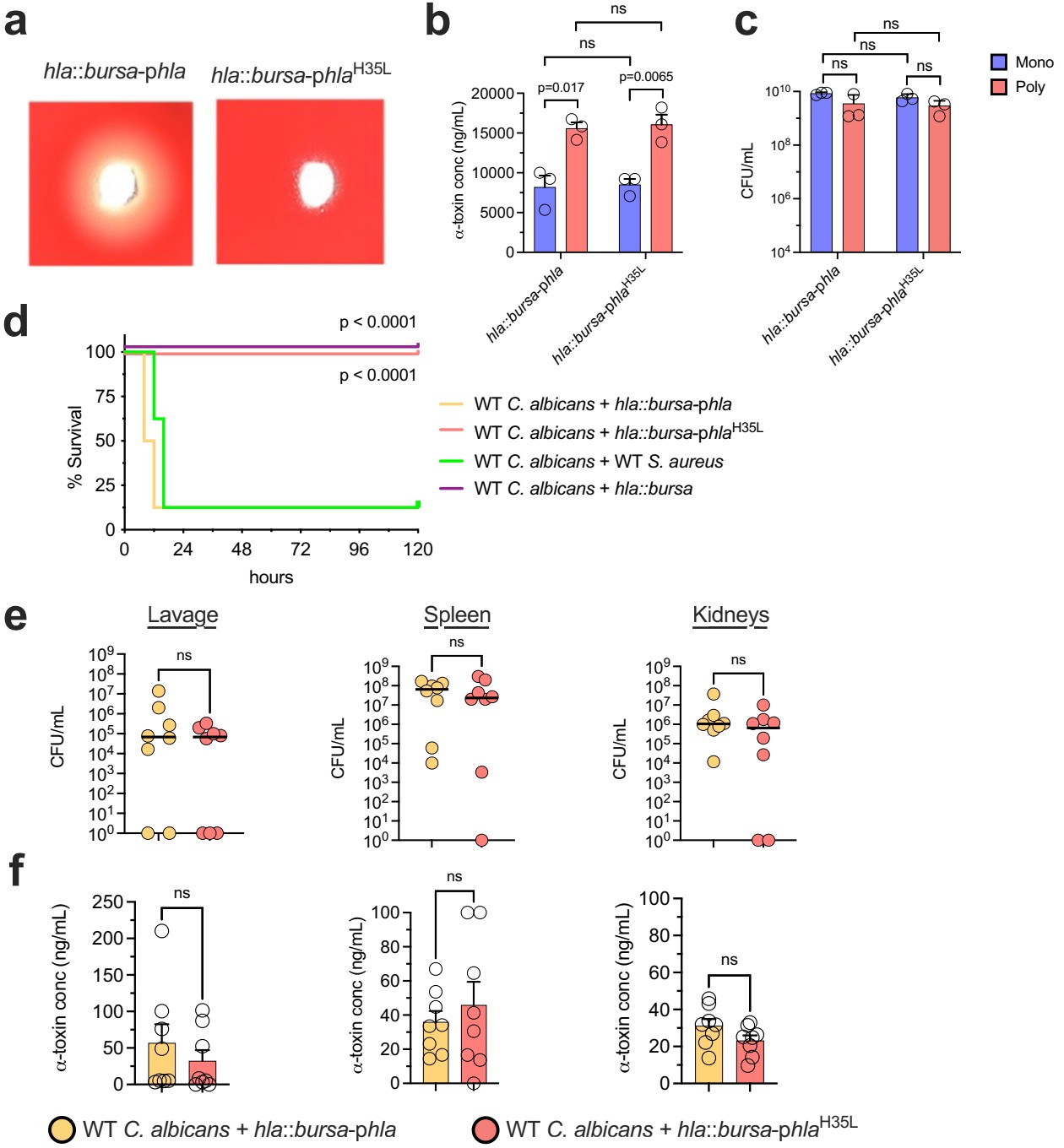

**Fig. 2 | α-toxin activity is required for lethal synergism during polymicrobial IAI. a** Hemolytic activity in filter-sterilized supernatants from *S. aureus hla*::*bursa*-p*hla* and *hla*::*bursa*-p*hla*^H35L cultures. Images are representative of independent (*n* = 3) experiments. **b** α-toxin from *S. aureus hla*::*bursa*-p*hla* and *hla*::*bursa*-p*hla*^H35L ± *C. albicans* (mono, blue; poly, pink) culture supernatants was measured via ELISA. Experiments were repeated in triplicate and expressed as mean + SEM. Significance was determined by two-sided Student's t-test. **c** Fungal and bacterial burdens were enumerated from experiments in panel b by selective plating. Experiments were repeated in triplicate and expressed as the mean + SEM. Significance was assessed by two-sided Student's t-test. ns, not significant. **d** Mice (*n* = 8 mice per group) were co-infected with WT *C. albicans* and WT *S. aureus*

(green line), *hla*::*bursa* (purple line), *hla*::*bursa*-p*hla* (orange line), or Δ*hla*-p*hla*^H35L (pink line). Survival was followed for up to 5 days post-infection (p.i.). Data are of two independent repeats of 4 mice per group and combined. Significance was assessed by comparing WT *C. albicans* + WT *S. aureus* with other groups using a two-sided Gehan-Breslow-Wilcoxon test. **e** Microbial burdens (*n* = 8 mice per group) were enumerated in the peritoneal lavage fluid, kidneys, and spleen. Line denotes the median. Significance was determined using a two-sided Mann–Whitney test. **f** α-toxin in the peritoneal lavage, kidneys, and spleen was measured by ELISA. Data is cumulative of two independent repeats and expressed as mean + SEM. Significance was determined by a two-sided Student's t-test. ns, not significant.

## Spatiotemporal induction of the *agr* quorum sensing system in vivo

To observe the spatiotemporal activation of the *agr* quorum sensing system in vivo during mono- and polymicrobial IAI, we constructed an

*agr* P3-luciferase reporter strain [*S. aureus*(pOLux)]. The *agr*-luciferase reporter responded comparably to the *agr*-GFP reporter, whereby signal increased ~2-fold during co-culture with WT *C. albicans* but failed to do so with *zcf13*Δ/Δ (Fig. 6a). Mice were then challenged IP

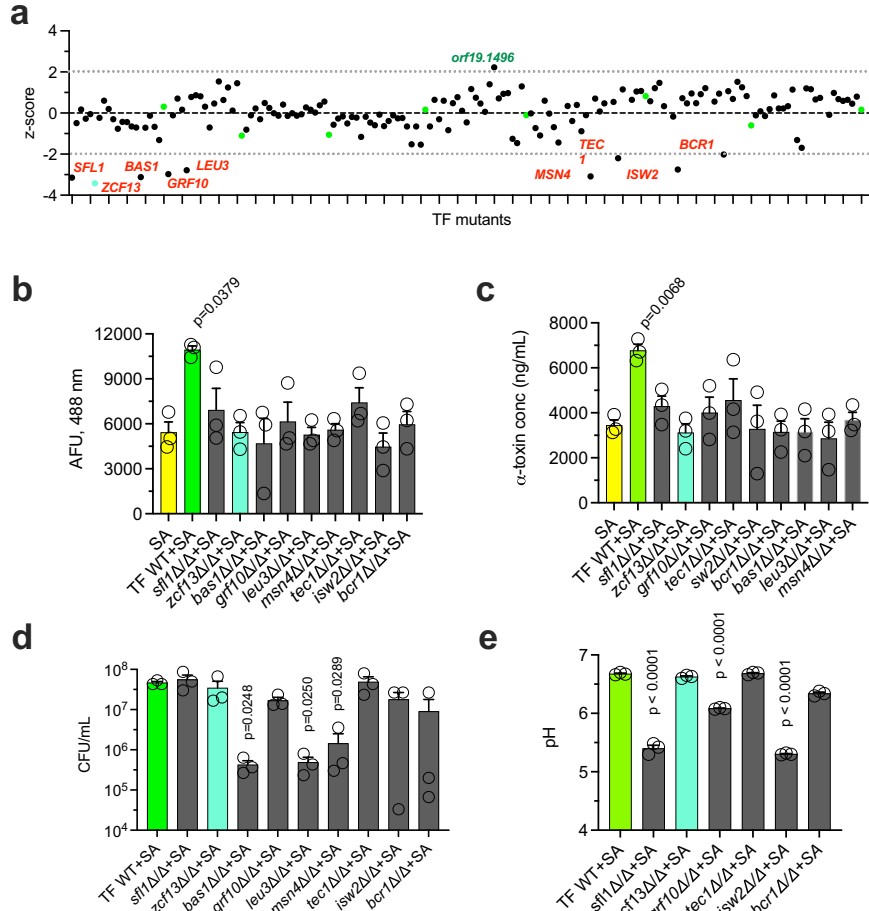

**Fig. 3 | Screen of *C. albicans* transcription factor mutants reveals novel regulators of *S. aureus agr* induction. a** TF WT (green dots) or mutants (black dots) were grown with *S. aureus*(pDB22) in co-culture. Fold-fluorescence was calculated and z-scores plotted. Mutants exhibiting >2-fold standard deviations from the population mean (dashed lines) are labeled in red (reduced) or green (enhanced). Mutants identified in **a** were re-confirmed or assessed for altered **b** *agr* enhancement (reporter assay), **c** α-toxin production (ELISA), **d** growth during co-culture (selective microbiological plating), and **e** culture pH (pH meter) at 16 h. All experiments (*n* = 3 biological replicates) are represented as the mean + SEM. Significance was assessed by comparing SA with other groups (**b**, **c**) and TF WT + SA with other groups (**d**, **e**) using a one-way ANOVA and Dunnet's post-test.

with *S. aureus*(pOLux) in the presence or absence of TF WT, TF *zcf13Δ/Δ*, SC WT, or SC *zcf13Δ/Δ C. albicans*. Luminescence was captured, and images were taken at 4 h intervals (Fig. 6b) and quantified (Fig. 6c). WT co-infections exhibited higher luminescence at each time point and demonstrated more dispersed *agr* signal in the abdomen, as compared to *S. aureus* mono-infection or co-infection with *zcf13Δ/Δ* mutant. This data further supports the hypothesis that *C. albicans* enhances *agr* activity during co-infection in a Zcf13p-dependent manner.

### Zcf13p regulates pentose sugar metabolism in *Candida*

Given its clear role in contributing to synergistic lethality in the peritoneal cavity, we next sought to understand the function of *ZCF13* in *C. albicans* by a transcriptional profiling approach. Among the differentially regulated metabolic genes observed, *zcf13Δ/Δ* showed strong down-regulation of *RBK1* that encodes for ribokinase involved in ribose catabolism (Fig. 7a, Supplementary Data 2)[42,43]. Phenotypic carbon substrate microarray experiments revealed that *zcf13Δ/Δ* had a significant growth defect as compared to SC5314 or *zcf13Δ/Δ+ZCF13* when D-ribose, D-arabinose, or 2-Deoxy-D-ribose (all pentose phosphate pathway (PPP) intermediates) were supplied as a sole carbon source (Fig. 7b, Supplementary Data 3). Decreased *RBK1* expression in *zcf13Δ/Δ* was confirmed by quantitative real-time PCR (qRT-PCR) during both mono- and co-culture as compared to WT and revertant strains (Fig. 7c, d). Collectively, these findings suggest the observed pentose

metabolism defect in *zcf13Δ/Δ* is partly due to reduced expression of *RBK1*.

### Ribose inhibits *S. aureus agr* quorum sensing

Prior studies demonstrated that ribose interferes with quorum sensing and biofilm formation in some bacterial species[44,45]. Thus, we wished to determine the impact of pentose sugars on *agr* quorum sensing in *S. aureus*. Therefore, *S. aureus*(pDB22) and *C. albicans* strains were co-cultured in TSBg supplemented with 20 mg/mL D-ribose, D-arabinose, and 2-Deoxy-D-ribose. While all pentose sugars showed inhibitory activity, ribose strongly attenuated *agr* activation during both mono- and co-culture (Fig. 8a). A dose-response study revealed that *agr* activation was suppressed in the presence of ≥0.2 mg/mL ribose, whereas *S. aureus* growth was suppressed at concentrations ≥1 mg/mL (Supplementary Fig. 3a, b). However, growth-normalized fluorescence values confirmed that ribose dose-dependently inhibited *agr* activation independent of its impact on growth (Fig. 8b). Yet, growth of SC5314 or *zcf13Δ/Δ* was not altered when cultivated in medium containing up to 20 mg/mL ribose (Supplementary Fig. 4a, b). Therefore, we utilized a non-growth inhibitory concentration of ribose (0.5 mg/mL) for subsequent experiments (confirmed in Fig. 8c). RNA-Seq was performed in the presence or absence of ribose to delineate the *S. aureus* transcriptional response to this pentose sugar (Supplementary Data 4 and 5). Transcriptional profiling showed that ribose

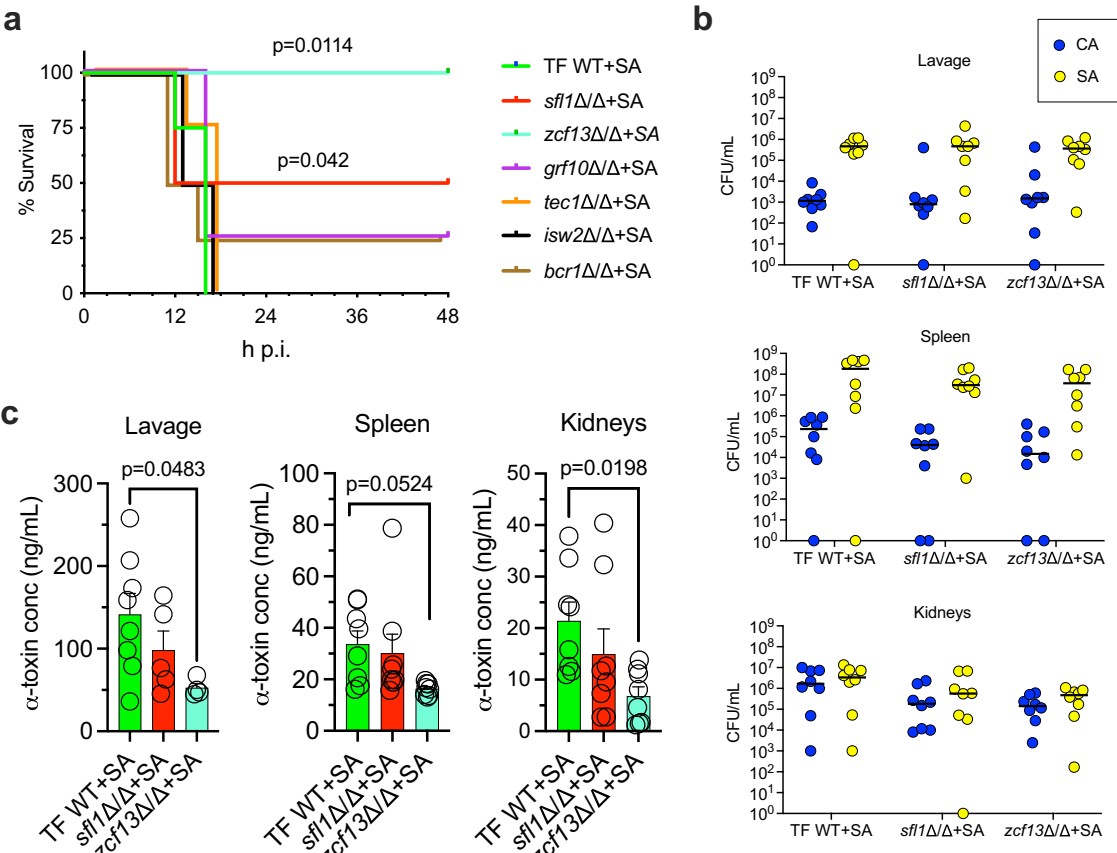

**Fig. 4 | *sfl1Δ/Δ* and *zcf13Δ/Δ* mutants display reduced virulence during poly-microbial infection. a** Mice (*n* = 8 per group) were inoculated with *S. aureus* (SA) and either TF WT (green) or select previously identified mutants (*sfl1Δ/Δ*, red; *zcf13Δ/Δ*, teal; *grf10Δ/Δ*, purple; *tec1Δ/Δ*, orange; *isw2Δ/Δ*, black; *bcr1Δ/Δ*, brown) and followed for survival. Experiments were repeated and data combined. Significance was assessed by comparing TF WT + SA with other groups using a two-sided Gehan-Breslow-Wilcoxon test. **b** Microbial burdens (*C. albicans*, blue; *S.* *aureus*, yellow) in the peritoneal lavage fluid, spleen, and kidneys (*n* = 8 mice per group) were enumerated at 8 h post-infection. Line represents the median. Significance was determined using a two-sided Mann–Whitney test. **c** α-toxin was measured in the peritoneal lavage fluid, spleen, and kidneys (*n* = 8 mice per group) by ELISA. Data is represented as mean + SEM. Significance was determined by comparing TF WT + SA with the other groups using a one-way ANOVA test.

inhibited the entire *agr* operon (*agrA, agrB, agrC, agrD*) and major effector of the quorum system (*hld/RNAIII*), consistent with our *agr* reporter data (Fig. 8d). Interestingly, the entire osmolarity responsive Kdp operon (encoded by *kdpABCDE*) was upregulated in the presence of ribose[46,47]. The DeoR family transcriptional regulator encoded by *fruR* was the highest differentially expressed gene induced by ribose and is homologous to ribose or glycerol-sensing proteins in other bacterial species[48,49]. The *purR* gene that encodes for a negative regulator of purine biosynthesis was increased in the presence of ribose, while its numerous downstream targets (e.g., *purC, purK, purQ, purS,* etc.) were concordantly repressed[50]. As ribose-5-phosphate is the major building block derived from the PPP for purine biosynthesis, excess exogenous ribose would be predicted to down-regulate purine anabolism.

### C. albicans Zcf13p-regulated ribose metabolism is essential for amplifying S. aureus agr activation and toxin production

Given that ribose is *agr* inhibitory and that Zcf13p is important for *C. albicans* ribose assimilation, we hypothesized that inefficient ribose metabolism by *zcf13Δ/Δ* may indirectly lead to repressed toxin production by *S. aureus* during co-culture. Therefore, *C. albicans* and *S. aureus* were mono- or co-cultured in TSBg or TSBg supplemented with 0.5 mg/mL ribose. WT and *ZCF13* revertant strains were able to drive *agr* activation above the *S. aureus* monomicrobial control (Fig. 9a). While some inhibition by ribose in these groups was noted, it was

mitigated presumably by the capacity of *C. albicans* to effectively metabolize the supplemented ribose. However, *zcf13Δ/Δ* was unable to elevate *agr* above the monomicrobial baseline and this was worsened with the addition of ribose (Fig. 9a). In support of this hypothesis, the concentration of ribose in spent media as measured by LC-MS was significantly lower in *S. aureus* co-cultured with SC5314 or *zcf13Δ/Δ* +*ZCF13* as compared to *zcf13Δ/Δ*, confirming a ribose metabolism defect in *zcf13Δ/Δ* (Fig. 9b). Importantly, ribose levels in vitro mirrored that of peritoneal lavage fluid obtained from uninfected mice, validating our experimental system (Supplementary Fig. 5). Indeed, an inversely proportional relationship was noted between the concentration of ribose and *agr* activation (Fig. 9c). Both qualitative and quantitative measures of toxin activity mirrored *agr* activation phenotypes (Fig. 9d, e). Remarkably, the concentration of ribose in the peritoneal lavage fluid of mice coinfected with *S. aureus* and *zcf13Δ/Δ* was significantly higher than that recovered from WT or revertant co-infections, confirming a ribose metabolism defect of *zcf13Δ/Δ* in vivo during murine IAI (Fig. 9f). Moreover, monomicrobial *S. aureus* infection did not significantly alter ribose concentration from baseline at this time point (Supplementary Fig. 5). Spent culture supernatants from WT or *zcf13Δ/Δ* mono- or co-cultures were filter sterilized and added to *S. aureus*(pDB22). Expressed as a ratio, the WT co-culture supernatant (ribose deplete) stimulated *agr* activation over that from the *zcf13Δ/Δ* (ribose replete) co-culture (Fig. 9g). Addition of monomicrobial supernatant had little to no effect. These findings confirm

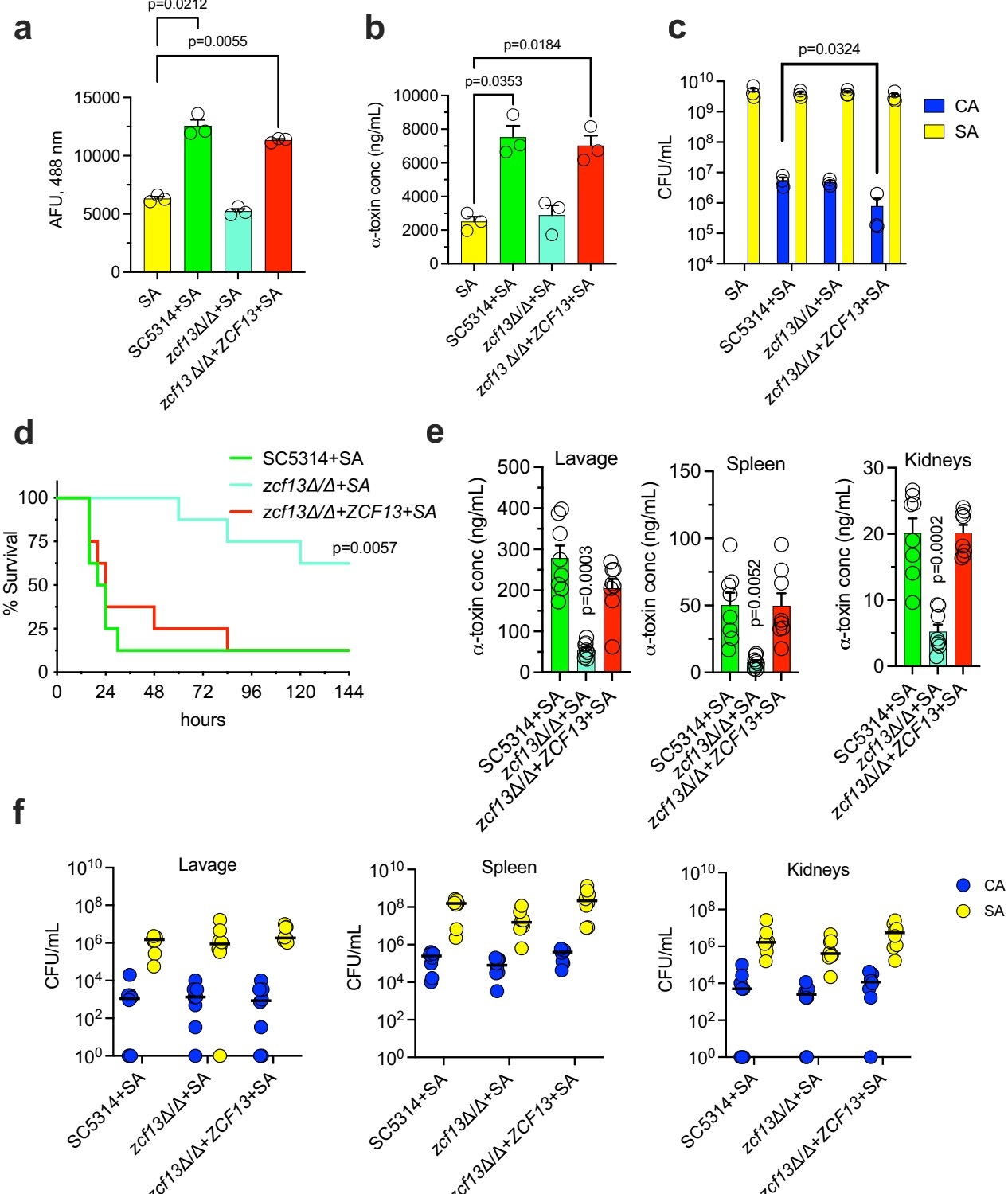

**Fig. 5 | Deletion of *ZCF13* in SC5314 abrogates *S. aureus* toxin production and synergistic lethality.** *S. aureus*(pDB22) was cultured alone (yellow) or with *C. albicans* SC5314 (green), *zcf13Δ/Δ* (teal), or *zcf13Δ/Δ+ZCF13* (red) in TSBg. Aliquots were removed to **a** measure fluorescence, **b** measure α-toxin by ELISA, or **c** enumerate microbial burden (CA, *C. albicans*; SA, *S. aureus*). Experimental data (*n* = 3) are expressed as mean + SEM. **d** Mice (n = 8 per group) were infected with SC5314, *zcf13Δ/Δ*, or *zcf13Δ/Δ+ZCF13* + *S. aureus* IP and monitored for survival for up to 5 d. Experiments were performed in duplicate and combined. Significance was assessed by comparing SC5314 + SA with the other groups. **e** α-toxin was measured by ELISA at 8 h post-infection (p.i.) in peritoneal lavage fluid, spleen, and kidneys from mice infected with the indicated strains. Data are cumulative of two independent repeats and represented as mean ± SEM. **f** Microbial burdens were enumerated at 8 h p.i. in lavage fluid, spleen, and kidneys of mice infected with the indicated strains were enumerated (*C. albicans*, blue; *S. aureus*, yellow). Data are cumulative of two independent repeats and expressed as the median. Significance was determined using the following tests: a one-way ANOVA and Dunnet's post-test (α-toxin), two-sided Mann–Whitney test (CFU), and two-sided Gehan-Breslow-Wilcoxon test (survival).

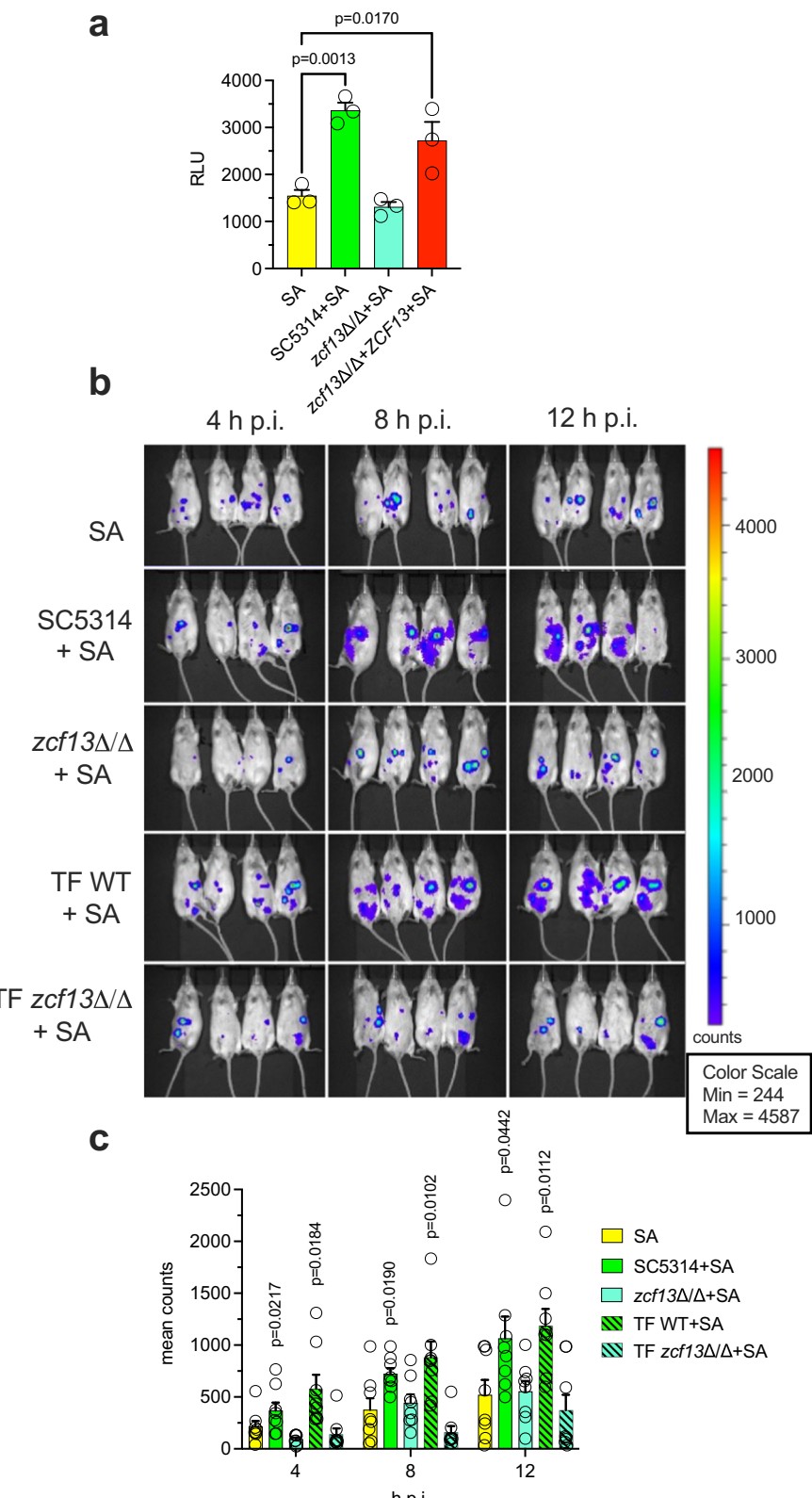

**Fig. 6 | Spatiotemporal *agr* activation during mono- and polymicrobial infection with WT and *zcf13Δ/Δ*. a** Luminescence was measured in mono- and polymicrobial cultures of *S. aureus* (pOLux) with or without *C. albicans* SC5314, *zcf13Δ/Δ*, *zcf13Δ/Δ+ZCF13* grown in TSBg. Data are cumulative of 3 independent experiments and expressed as mean + SEM. Significance was determined by comparing SA with the other groups using a one-way ANOVA and Dunnett's post-test. **b** Mice were infected with strains described in (**a**). Images were taken every 4 h p.i. using a Xenogen IVIS Spectrum. Images are uniformly scaled. **c** Luminescence values ($n$ = 8 mice per group) were quantified within regions of interest and plotted as mean + SEM. Significance was assessed by comparing SA with the other groups using a two-sided Mann–Whitney test to compare isogenic strain sets.

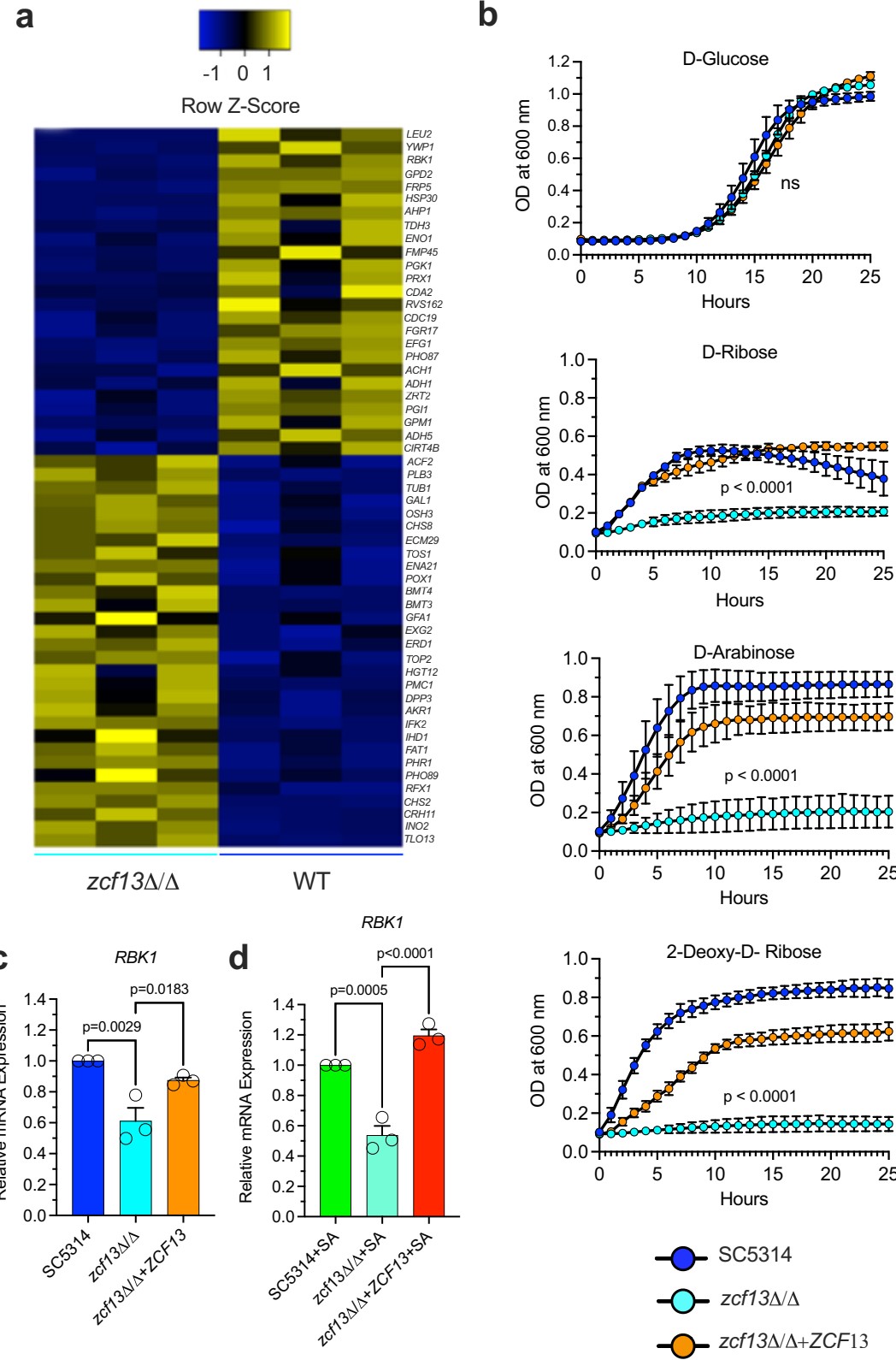

**Fig. 7 | *ZCF13* plays a role in ribose and pentose sugar metabolism. a** WT *C. albicans* and *zcf13Δ/Δ* were grown in TSBg and transcriptional profiling was performed by RNA-Seq. Heatmap depicting significantly differentially expressed genes (blue, decreased; yellow, increased) showing ≥1.5-fold changes (*p* < 0.05 Student's t-test, FDR < 0.01). **b** SC5314 (blue), *zcf13Δ/Δ* (cyan), and *zcf13Δ/Δ+ZCF13* (orange) were cultured in Biolog carbon source plates containing growth medium that otherwise lacked a carbon source. Growth was kinetically monitored by OD600 nm. *RBK1* expression from transcriptional profiling was validated by qRT-PCR in both **c** mono-culture and **d** co-culture. All experiments were repeated in biological triplicate and shown as mean + SEM. Significance was assessed by comparing SC5314 or SC531 + SA with the other groups using one-way ANOVA and Dunnett's post-test. ns, not significant.

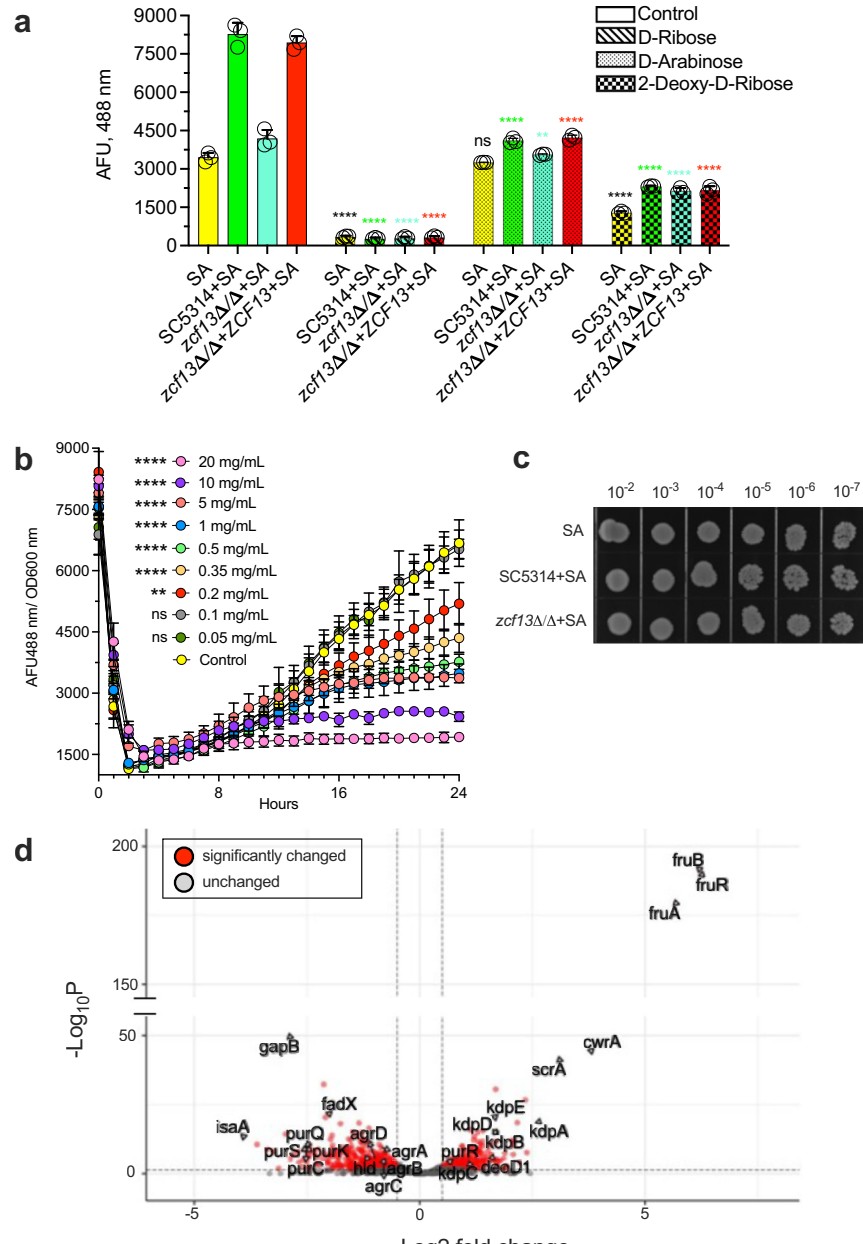

**Fig. 8 | Ribose suppresses *S. aureus agr* activation and growth. a** *S. aureus*(pDB22) was grown without (yellow) or with SC5314 (green), *zcf13Δ/Δ* (teal), and *zcf13Δ/Δ+ZCF13* (red) in TSBg (open) or TSBg supplemented with 2% pentose sugars [ribose (diagonals), arabinose (dots), 2-deoxy-D-ribose (checkered)] and fluorescence measured. Results (*n* = 3 biological replicates) are shown as mean + SEM. Significance was assessed using a one-way ANOVA and Dunnett's post-test. **\****p* = 0.0072; \*\*\**p* < 0.001; ns, not significant. **b** *S. aureus* was grown in TSBg or that supplemented with different concentrations of ribose (*n* = 3 biological replicates) as indicated. Kinetic fluorescence data (488 nm) was normalized by OD600 nm values. All groups were compared to the ribose-free control. Significance was assessed using a one-way ANOVA and Dunnett's multiple comparisons post-test. **\****p* = 0.0036; \*\*\**p* < 0.0001. **c** *S. aureus*(pDB22) was co-cultured with SC5314, *zcf13Δ/Δ*, or *zcf13Δ/Δ+ZCF13* in TSBg supplemented with 0.5 mg/mL ribose. Growth was assessed by plating on TSB containing 10 μg/mL amphotericin B. Representative images (*n* = 3 replicates) are depicted. **d** *S. aureus* was grown in TSBg with (0.5 mg/mL) and without ribose (*n* = 4 replicates) and transcriptional profiling performed by RNA-Seq. Volcano plot depicting a subset of differentially expressed genes (red) showing ≥1.5-fold changes that were considered significant (p < 0.05, two-sided Student's t-test, FDR < 0.01).

that a soluble factor phenocopies *agr* activation observed during live co-culture. Taken together, these findings demonstrate that fungal ribose catabolism is critical for *C. albicans*-induced *agr* activation and toxin production in *S. aureus*.

### *RBK1* and *HGT7* expression are controlled by *ZCF13* and are essential for robust *S. aureus agr* activation

To determine whether *RBK1* (a putative downstream target of Zcf13p) impacted *agr* activation during co-culture, *rbk1Δ/Δ* and *rbk1Δ/Δ+RBK1* strains were constructed as described (Supplementary Methods)[51,52]. Similar to *zcf13Δ/Δ*, the *rbk1Δ/Δ* mutant led to reduced *agr* activation during co-culture and this was reverted to WT levels with the *rbk1Δ/Δ +RBK1* strain (Fig. 10a). As expected, *RBK1* deletion did not impact *ZCF13* expression (Supplementary Fig. 6). Co-infection of *rbk1Δ/Δ* with *S. aureus* failed to drive early infectious synergism and mice survived significantly longer as compared to co-infection with SC5314 or *rbk1Δ/Δ+RBK1* (Fig. 10b). While *C. albicans* readily metabolizes ribose under co-culture conditions and *RBK1* plays a key role in this process, the

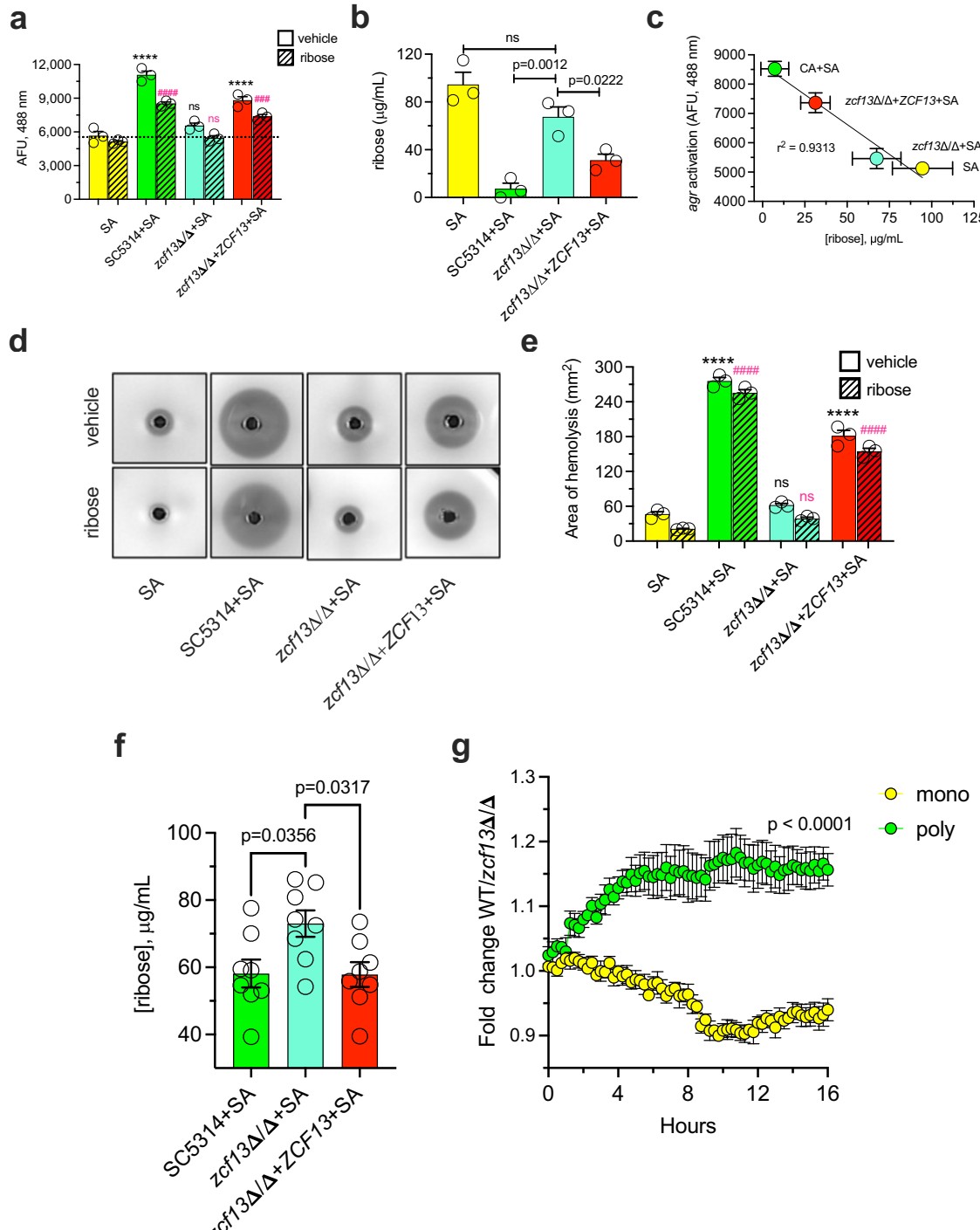

**Fig. 9 | *C. albicans* ribose metabolism derepresses staphylococcal *agr* activation and α-toxin production.** *S. aureus*(pDB22) was grown in the absence (yellow) or presence of SC5314 (green), *zcf13Δ/Δ* (teal), and *zcf13Δ/Δ+ZCF13* (red) in TSBg containing 0.5 mg/mL ribose for 20 h and **a** *agr* activation (dashed line depicts monomicrobial baseline) or **b** ribose concentrations in filter-sterilized supernatant was measured. **c** Correlation between the concentration of ribose and *agr* activation was determined using linear regression analysis. The activity of α-toxin in mono and co-culture supernatant was **d** qualitatively and **e** quantitatively assessed. Experiments were repeated in biological triplicate and shown as mean + SEM. Significance was assessed by comparing SA with the other groups in absence or presence of ribose using one-way ANOVA, Tukey's or Dunnett's post-tests. $^{###}p = 0.003$; **** or $^{####}p < 0.0001$; ns, not significant. **f** Mice ($n = 8$ per group) were coinfected with *S. aureus* and indicated *C. albicans* strains. The concentration of ribose at 8 h p.i. in peritoneal lavage fluid was measured using LC-MS. Data are depicted as the mean ± SEM. Significance was assessed using one-way ANOVA and Tukey's post-test. **g** *S. aureus*(pDB22) was grown in TSBg supplemented with filter-sterilized SC5314 or *zcf13Δ/Δ* mono-culture (yellow) or co-culture (green) spent medium. Activation of *agr* was kinetically monitored and expressed as fold-change of WT over *zcf13Δ/Δ*. Data ($n = 4$ biological replicates) are presented as mean ± SEM. Significance was assessed by a two-sided multiple Student's t-test.

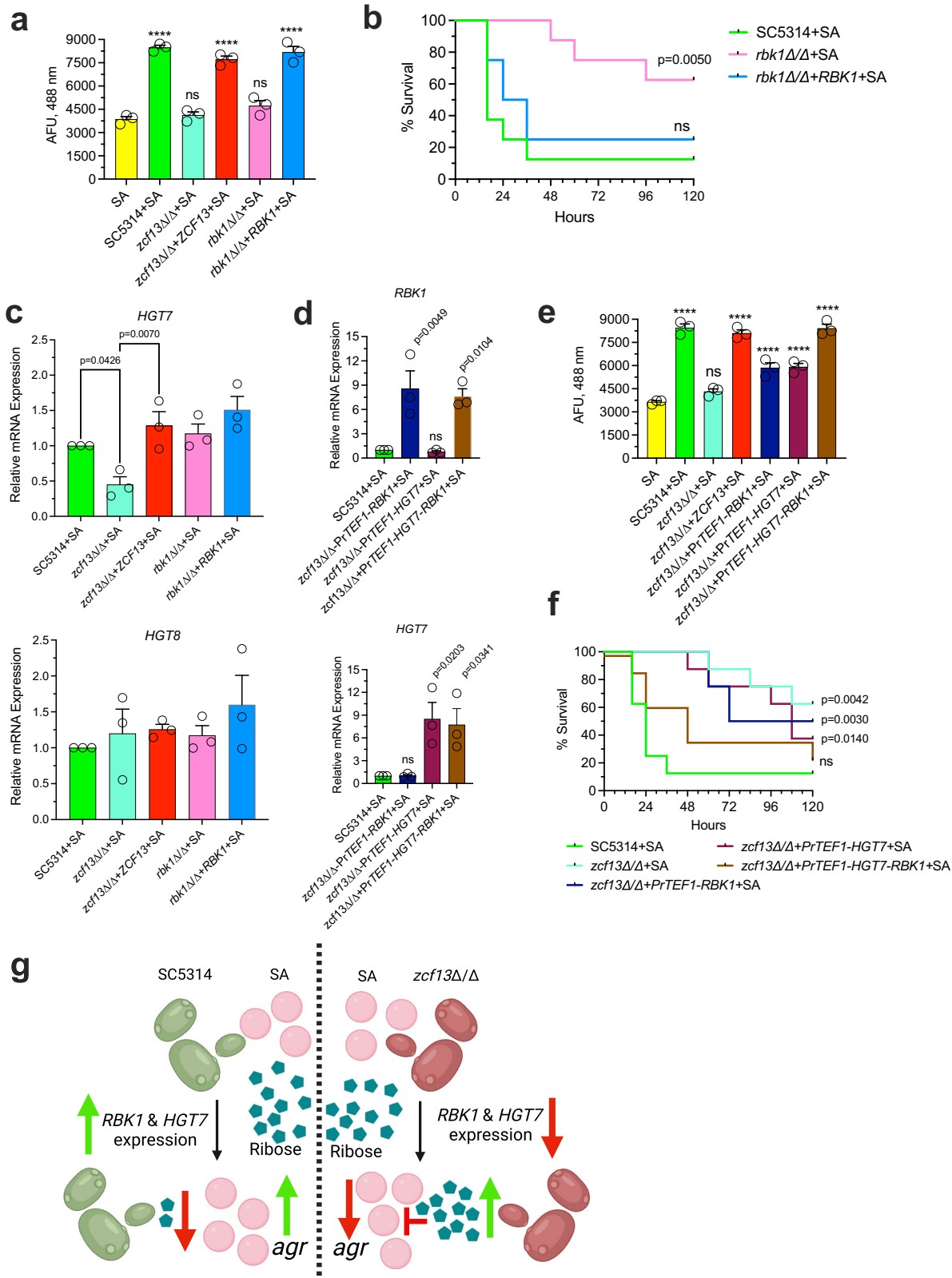

transcriter associated with ribose import is unknown. It was reported that *GAL2* in *Saccharomyces cerevisiae* acts as a low-affinity ribose importer[53]. A homology search revealed that the *C. albicans* sugar importers CaHgt7p (57.4% identity) and CaHgt8p (59.4% identity) share conserved amino acid sequence with ScGal2p (Supplementary Fig. 7). The expression of *HGT7*, but not *HGT8*, was significantly lower in *zcf13*Δ/Δ during co-culture when compared to SC5314 (or other

constructed strains), confirming that expression of *HGT7* is partially *ZCF13*-dependent (Fig. 10c). Therefore, strains with forced expression of *RBK1*, *HGT7*, or both genes were constructed in *zcf13*Δ/Δ and their expression confirmed by qRT-PCR (Fig. 10d). Overexpression of *RBK1* or *HGT7* was able to partially revert the *agr* activation phenotype when co-cultured with *S. aureus* as compared to WT *C. albicans*. However, simultaneous overexpression of *RBK1* and *HGT7* fully restored *agr*

**Fig. 10 | *ZCF13*-regulated *RBK1* and *HGT7* expression is crucial for augmented *agr* activation in *S. aureus* during co-culture and co-infection. a** SC5314 (green), *zcf13Δ/Δ* (teal), *zcf13Δ/Δ+ZCF13* (red), *rbk1Δ/Δ* (pink) and *rbk1Δ/Δ+RBK1* (blue) were grown with *S. aureus*(pDB22) in TSBg and fluorescence measured. **b** Mice (*n* = 8 per group) were coinfected with SC5314, *rbk1Δ/Δ*, or *rbk1Δ/Δ+RBK1* and *S. aureus* IP and monitored for survival for up to 5 days. Significance was assessed by comparing SC5314 + SA with other groups using a two-sided Gehan-Breslow-Wilcoxon test. **c** The indicated strains were cultured with *S. aureus*(pDB22) in TSBg and expression of *HGT7* and *HGT8* was measured by qRT-PCR using the $2^{-\Delta\Delta Ct}$ method normalizing to SC5314 + SA and *ACT1*. **d** *S. aureus*(pDB22) was co-cultured with SC5314 (green), *zcf13Δ/Δ+PrTEF1-RBK1* (navy)*, zcf13Δ/Δ+PrTEF1-HGT7* (plum) & *zcf13Δ/Δ+PrTEF1-HGT7-RBK1* (brown) in TSBg and expression of *RBK1 and HGT7* measured as

described above. **e** Similar to panel (**a**), *agr* activation was determined in the indicated strains. All in vitro experiments (*n* = 3 biological replicates) are shown as mean + SEM. Significance was assessed using one-way ANOVA and Dunnett's post-test. ****$p$ < 0.0001; ns, not significant. **f** Mice (*n* = 8 per group) were coinfected with SC5314, *zcf13Δ/Δ*, *rbk1Δ/Δ*, *zcf13Δ/Δ+PrTEF1-RBK1*, *zcf13Δ/Δ+PrTEF1-HGT7,* or *zcf13Δ/Δ+PrTEF1-HGT7-RBK1* and *S. aureus* IP and monitored for survival. Significance was assessed by comparing SC5314 + SA with the other groups using a two-sided Gehan-Breslow-Wilcoxon test. **g** Working model of *C. albicans* ribose metabolism impacting staphylococcal *agr*-mediated toxicity. Created with Bior-ender.com released under a Creative Commons Attribution-NonCommercial-NoDerivs 4.0 International license (https://creativecommons.org/licenses/by-nc-nd/4.0/deed.en).

activation similar to co-culture with SC5314 (Fig. 10e). To determine whether simultaneous expression of *RBK1* and *HGT7* is crucial for infectious synergism, mice were co-infected with these isogenic strains. Similar to in vitro observations, co-infection with the *RBK1* or *HGT7* single overexpression strains only modestly increased overall mortality (Fig. 10f). However, co-infection with the *RBK1-HGT7* double overexpression strain showed significantly greater mortality that was similar to WT co-infection (Fig. 10f). These findings confirmed that *RBK1* and *HGT7* are at least partly under the control of Zcf13p and that their coordinated expression is crucial for fully enhanced staphylococcal *agr* activation and infectious synergism during co-infection with *C. albicans*.

Thus, our experimental data supports the following working model as depicted in Fig. 10g. During co-culture or co-infection, the *C. albicans* Zcf13p transcription factor is activated and upregulates downstream target genes *RBK1* and *HGT7*, leading to efficient ribose uptake and metabolism via the PPP. As ribose (or other potential pentose sugar substrates) are depleted from the environment, the *S. aureus agr* quorum system is derepressed to drive high levels of toxin and lethality during IAI. In the case of impaired *C. albicans* ribose catabolism (as observed with *zcf13Δ/Δ*), exogenous ribose concentrations remain elevated and lead to extended inhibition of staphylococcal quorum signals and toxins.

## Discussion

Despite their clear contribution to shaping human disease, polymicrobial infections remain generally understudied and the mechanistic impact of ensuing microbe-microbe interactions are poorly understood. This is not surprising given the complexities of microbial composition, diverse host niches, and biologically relevant experimental modeling. Dynamic regulation of host responses to multiple pathogens, which are simultaneously responsive to fluctuating environmental conditions, adds additional layers of complexity.

Despite these challenges, our laboratories have expended concerted effort to unravel the mechanisms that drive synergistic lethality during intra-abdominal infection (IAI) with *C. albicans* and *S. aureus*, including defining contributions of host inflammation, trained innate immunity, and microbial virulence determinants, including a key requirement for staphylococcal α-toxin[28,54–58]. The function of α-toxin is multifactorial, possessing the capacity to induce hemolysis, drive pro-inflammatory responses, and disrupt platelet function. During intravenous delivery of *S. aureus* or purified α-toxin, Surewaard et al. observed significant platelet aggregation in the livers of infected mice that was absent when a Δ*hla* mutant, H35L α-toxin variant, or passive immunization with the neutralizing antibody MEDI4893* was employed[30]. During WT as compared to Δ*hla* infection, circulating platelets were notably decreased, indicating a role for α-toxin in inducing thrombocytopenia. Additionally, the liver enzyme ALT and focal necrotic lesions were decreased during infection with Δ*hla*. An additional study conducted by Powers et al. demonstrated that α-toxin alters platelet activation as well as promotes the formation of platelet-neutrophil aggregates, which contribute to lung and liver damage

during staphylococcal sepsis[29]. These studies mirror our own findings during peritoneal infection, where organ damage biomarkers were elevated during co-infection and dependent on oligomerization-sufficient α-toxin. In addition to altered platelet activity, *S. aureus* has been described as a "master manipulator" of host clotting cascades and is equipped with several coagulases and agglutinins that promote noncanonical coagulation or facilitate binding of fibrin and fibrinogen[59]. These factors could contribute to exacerbated coagulation during co-infection beyond those mediated via α-toxin. In either case, it is likely that organ damage due to ensuing coagulopathy plays a crucial role in deleterious outcomes observed during polymicrobial IAI and eventual sepsis. In fact, onset of clinical disseminated intravascular coagulation (DIC), resulting in body-wide coagulation, eventual consumption of endogenous clotting factors, and organ dysfunction, doubles the sepsis mortality rate[60–62]. Additional studies to define contributions of coagulation during polymicrobial IAI are ongoing in our laboratories.

Given its pivotal role during murine IAI, an unbiased screening approach was employed to identify fungal transcriptional regulators responsible for the previously reported upregulation of staphylococcal α-toxin during co-culture and co-infection to better understand how *C. albicans* potentiates *S. aureus* virulence[12,28]. While several transcription factor mutants were identified, only *zcf13Δ/Δ* failed to drive elevated α-toxin levels without noted growth or external alkalinization defects. While a complete picture of Zcf13p function remains unclear, prior studies have reported disparate phenotypes for *zcf13Δ/Δ*. In the seminal study which reported construction of the transcription factor library employed here, no temperature susceptibility, chemical sensitivity, or morphological abnormalities were noted[36]. Using a pooled-infection approach, Vandeputte, et al. showed that a *zcf13-1* disruption mutant (created by insertional mutagenesis) displayed reduced colonization of the murine kidney and subsequent single-strain infections confirmed this phenotype[63]. They also noted susceptibility of *zcf13-1* to elevated temperature and increased colony wrinkling and agar invasion. However, these aberrant phenotypes were shown to be associated with unintended Tn7-driven transcripts produced as a result of the insertional mutagenesis strategy used[64]. Subsequent work using a complete *zcf13Δ/Δ* deletion mutant and revertant by Amorim-Vaz et al. failed to find any murine kidney fungal burden or virulence defects, which are reflective of similar fitness to WT observed in our IAI model[65].

Despite no obvious growth or virulence defect associated with the loss of *ZCF13* during monomicrobial infection, here we show its function is at least partly associated with metabolism, including utilization of the pentose sugar D-ribose, and it is required for driving lethal synergism during *S. aureus* co-infection. Ribose plays critical roles for the cellular manufacturing of nucleotides, key biochemical redox electron acceptors (e.g., flavin adenine dinucleotide [FAD] and nicotinamide adenine dinucleotide/phosphate [NAD and NADP]), some amino acids, and signaling molecules (e.g., cyclic adenosine monophosphate [cAMP])[66]. It can be synthesized by glycolytic flux via the pentose phosphate pathway (PPP), whereby the oxidative branch leads

to the production of ribose-5-phosphate that can be further converted via the non-oxidative branch to glycolytic end products[66]. Ribose may also be directly imported from exogenous sources and phosphorylated by ribokinase (encoded by *RBK1*) to ribose-5-phosphate that is then committed to the PPP[67]. In agreement with our phenotype microarray results, *C. albicans* (and *S. cerevisiae*) was previously shown to catabolize ribose completely as a sole carbon source, suggesting it can be taken up from the environment[68]. Presumably, this is driven by the ScGal2p low affinity ribose importer ortholog encoded by *HGT7* interrogated here[53]. However, this does not rule out that other sugar transporters may contribute to ribose import in *C. albicans*. As our results demonstrate partial restoration of lethal synergism with *RBK1* or *HGT7* overexpression in the *zcf13Δ/Δ* mutant but nearly WT-like synergism when co-overexpressed, this suggests coordinated regulation of ribose catabolism under the control of Zcf13p. This would be functionally similar to well-described ribose uptake and utilization mechanisms in *E. coli* and other bacteria that are carried out by a ribose operon regulator RbsR (i.e., CaZcf13p), a ribokinase RbsK (i.e., CaRbk1p), a ribose importer RbsABC/RbsU (i.e., CaHgt7p), and a ribose pyranase RbsD[69,70]. While RbsD homologs do not exist in yeast, it is conceivable that an incompletely characterized isomerase could convert ribose between its pyranose and furanose forms or that this activity is dispensable for Rbk1p-mediated ribose metabolism in *C. albicans*[71]. Interestingly, robust catabolism of ribose by *C. albicans* only occurred during co-culture or when it was supplied as a sole carbon source, hinting that carbon catabolite repression by other preferred carbon sources may impede its uptake as described in *S. cerevisiae*[72,73]. While there is a clear role for ribose impacting *agr* regulation, we cannot rule out the possibility that divergent carbon sources in vivo may be alternatively metabolized by *C. albicans* or directly interfere with *agr* signaling during co-infection. Unbiased metabolomic profiling of monomicrobial and co-infection may elucidate such additional contributory factors.

Transcriptional analysis revealed a clear repressive effect of ribose on the *agr* quorum sensing system that underpins toxicity of *S. aureus*, which is in line with the inability of *zcf13Δ/Δ* to deplete ribose levels during co-culture resulting in repressed toxin production[74]. While ribose has been shown to competitively inhibit autoinducer-2 (AI2) quorum sensing and biofilm growth in *Haemophilus*, *Actinobacillus*, and *Lactobacillus*, the LuxS/AI2 circuit in *S. aureus* is instead associated with capsular polysaccharide (CP) production via the KdpDE regulon[44,75–77]. Interestingly, deletion of *luxS* that drives production of AI2 (which would functionally phenocopy ribose-mediated inhibition) leads to increased CP synthesis in *S. aureus* and we noted upregulation of the entire Kdp operon in our dataset[77]. However, given that AI2 is decoupled from toxin regulation in *S. aureus*, ribose is unlikely to directly competitively inhibit toxicity via a canonical receptor mechanism[78]. This begs the question as to how ribose exerts its effect on toxin regulation. There are two possibilities to explain our observations. Firstly, elegant work by the Somerville and Torres laboratories have revealed that the staphylococcal transcriptional repressor RpiRc can sense intracellular biosynthetic intermediates as a gauge of carbon flow through the PPP to repress virulence[79,80]. In fact, independent *ΔrpiRc* mutants demonstrated increased *agr* activation, *hla* transcription, leukocidin (LukSF-PV and LukED) production, and hemolysis. Thus, sensing of ribose levels directly (or other PPP intermediate(s)) via RpiRc is one viable mechanism by which *S. aureus* is responding to environmental changes elicited by *C. albicans* during co-culture. Another possibility is that the staphylococcal purine biosynthesis regulator PurR may be implicated in modulating virulence during growth with *C. albicans*[81,82]. Ribose-5-phosphate can serve as the primer for the synthesis of purine nucleotides and under ribose replete conditions purine synthesis genes are repressed by PurR as the starting material is plentiful. As ribose levels are depleted, purine synthesis is derepressed. In addition to dysregulated purine biosynthesis, a study by Sause et al. revealed that a *S. aureus* Δ*purR* mutant exhibited elevated levels of exoproteins, including toxins, and exacerbated virulence in a murine model of systemic infection[50]. Our transcriptional dataset reported here demonstrated that exogenous ribose repressed the entire staphylococcal purine biosynthetic pathway and increased PurR transcription. Thus, as ribose levels decrease during co-culture and co-infection, inactivated PurR may also contribute to hypervirulence of *S. aureus* in the presence of *C. albicans*. However, additional work to delineate the contributions of these pathways remains an active area of interest.

Given that ribose is estimated to be the second most abundant carbohydrate in human serum (after glucose), our findings may be highly relevant to other clinical polymicrobial infections of the abdominal cavity or extend to other systemic or mucosal sites[83]. Collectively, the work presented here reinforces the important conceptual intersection of metabolism and virulence[10]. Through alteration of the metabolic landscape, a common fungal commensal can augment the pathogenicity of a significant opportunistic pathogen with devastating consequences for the host. These studies may inform the design of non-metabolizable substrate analogs to dysregulate carbon sensing and the production of critical bacterial virulence determinants to improve unacceptably poor outcomes associated with polymicrobial sepsis.

## Methods

### Ethics statement

The animals used in this study were housed in AAALAC-approved facilities located at the University of Tennessee Health Sciences Center (UTHSC) in the Regional Biocontainment Laboratory (RBL). The UTHSC Animal Care and Use Committee, Laboratory Animal Care Unit (LACU) approved all animal usage and protocols (protocol #18-060 and 21-0266.0). Mice were maintained at ambient temperature (65–75 °C) and humidity (40–60%) under 12 h dark/light cycles. Mice were given standard rodent chow and water ad libitum. Mice were monitored daily for signs of distress, including noticeable weight loss and lethargy and humane endpoints used. UTHSC LACU uses the Public Health Policy on Humane Care and Use of Laboratory Animals (PHS) and the Guide for the Care and Use of Laboratory Animals as a basis for establishing and maintaining an institutional program for activities involving animals. To ensure high standards for animal welfare, UTHSC LACU remains compliant with all applicable provisions of the Animal Welfare Act (AWAR), guidance from the Office of Laboratory Animal Welfare (OLAW), and the American Veterinary Medical Association Guidelines on Euthanasia.

### Strains and growth conditions

Bacterial and fungal strain details can be found in Supplementary Tables 1 and 2. *Candida albicans* strain SC5314 (CA) was used as the wild-type/reference strain for all experiments unless otherwise noted. A library of *C. albicans* transcription factor deletion mutants and accompanying wild-type background strain (TF WT) were obtained from the Fungal Genetics Stock Center[36]. Strains were maintained as glycerol stocks and stored at −80 °C. *S. aureus* strain JE2 (SA) (a USA300 isolate used as wild-type) and strain NE1354 (*hla::bursa*, α-toxin-deficient) were obtained from the Biodefence and Emerging Infectious (BEI) Research Resources repository[84]. A *S. aureus* reporter strain [*S. aureus*(pDB22)] (plasmid containing the P3 promoter fused to GFP_mut2 and erythromycin resistance cassette) was also used in this work, as described previously[37]. Newly created plasmids, bacterial strains, and fungal strains used throughout this study were constructed by standard or published protocols and common phenotypes assessed as detailed in the Supplementary Methods. *Candida* strains were streaked onto yeast-peptone-dextrose (YPD) agar plates and grown at 30 °C. Single colonies were inoculated into 1.5 mL YPD broth and grown at 30 °C with shaking at 200 rpm. *S. aureus* strains were

streaked onto trypticase soy agar (TSA) (with antibiotics added as needed) and grown at 37 °C. Single colonies were inoculated into 1.5 mL TSB (with antibiotic added as needed) and grown at 37 °C with shaking at 200 rpm. *Escherichia coli* strains DH5-α and IM08B (obtained through BEI Resources) were used for plasmid construction and were grown on Luria-Bertani (LB) agar supplemented with 100 μg/mL ampicillin or 50 μg/mL kanamycin as described[85,86].

## Murine model of intra-abdominal infection

The mouse model of polymicrobial IAI was conducted as described previously[26,28,54–56]. As sex does not impact outcome, groups ($n = 4$–8) of 6–8-week-old female Swiss Webster mice were injected intraperitoneally (IP) with $1.75 \times 10^7$ CFU of *C. albicans*, $8 \times 10^7$ CFU of *S. aureus*, or $1.75 \times 10^7$ and $8 \times 10^7$ CFU of each microbe simultaneously. Inocula were prepared in a final volume of 0.2 mL pyrogen-free phosphate-buffered saline (PBS). After inoculation, mice were observed up to 5 d for morbidity (hunched posture, inactivity, ruffled fur) and mortality. In some experiments, experiments were terminated at 8 h p.i. Peritoneal cavities were lavaged by injection of 2 mL of sterile PBS containing 1X cOmplete protease inhibitors (Roche, Cat. No. 11836153001) or stable isotope labeling with amino acids in cell culture (SILAC) RPMI (for mass spectrometry) followed by gentle massaging of the peritoneal cavity. Peritoneal lavage fluid was then removed using a pipette inserted into a small incision in the abdominal cavity. Both kidneys and the spleen were removed from infected mice and placed in 500 uL PBS for homogenization prior to CFU enumeration and ELISA analysis. Whole blood was collected by cardiac puncture, and serum was separated by centrifugation. Animal experiments were repeated in duplicate and results combined.

## Clinical chemistry analysis

Clinical chemistry analysis of serum was performed using a DiaSys Respons® 910Vet chemistry analyzer (DiaSys Diagnostic Systems, USA, Wixon, MI). All tests were calibrated (TruCalU calibrator, DiaSys Diagnostic System), and bi-level quality control materials (TruLab N and TruLab P, DiaSys Diagnostic Systems) were run prior to sample analysis. The Respons®910VET chemistry analyzer uses colorimetry with either a rate or end-point reaction method. All reagents were purchased from DiaSys and analyses performed by RBL staff as fee-for-service according to the manufacturer's established procedures. Data is presented as the mean + standard error of the mean (SEM).

## IVIS imaging

Groups of mice ($n = 4$) were infected, as described above, with *S. aureus*(pOLux) (see Supplementary Methods). To minimize background interference, mice were given alfalfa-free rodent chow (Envigo) for 1 week prior to infection and imaging. Bioluminescence imaging was performed using a Xenogen IVIS Spectrum. Mice were anesthetized with isoflurane and imaged at 2–4 h intervals[87]. Images are uniformly scaled and average counts within regions of interest were determined with Living Image 4.7.3. Data is presented as the mean + SEM.

## GFP and luciferase *agr* reporter assays

Overnight cultures of *C. albicans* (YPD at 30 °C) and *S. aureus*(pDB22) or *S. aureus*(pOLux) (TSB at 37 °C) were washed three times with phosphate-buffered saline (PBS) by centrifugation. Cell concentrations were adjusted to $1 \times 10^5$ cells/mL in 5 mL of 0.6x TSB + 0.2% glucose (TSBg) for monomicrobial (CA or SA) or polymicrobial (CA + SA) cultures. In some experiments, 2% of various pentose sugars were added as indicated. Antibiotic (10 μg/mL erythromycin or 10 μg/mL chloramphenicol) was added for plasmid maintenance. Cultures were incubated at 37 °C with shaking at 200 rpm for 16 h. 100 μL aliquots were removed in triplicate and added to wells of black (fluorescence) or white (luminescence) 96-well microtiter plates.

Fluorescence (488 nm excitation, 525 nm emission) or luminescence (integration time 1 min) was measured using a Synergy H1 plate reader (Biotek). In some experiments, *S. aureus*(pDB22) was incubated in TSBg supplemented with an equal volume of spent culture supernatant from WT or *zcf13*Δ/Δ mono- or co-cultures. Experiments were repeated in triplicate and expressed as mean arbitrary fluorescence units (AFU) or relative light units (RLU) + SEM.

## Blood agar lysis assay

Mono- and polymicrobial cultures were prepared as above. At 16 h post-inoculation, 5 mL of culture was centrifuged at $2400 \times g$, and the resulting supernatant sterilized using a 0.2 μm syringe filter. To evaluate hemolysis, 400 μL sterile supernatant was precipitated with 1.6 mL chilled acetone and resuspended in 30 μL sterile growth media. Holes were made in blood agar plates (TSA with 5% sheep's blood) using a sterile pipette tip. Concentrated supernatants after precipitation were added to the wells, and the plates were incubated at 37 °C for 24 h. Plates were imaged with a digital scanner (EPSON Perfection V700 Photo) or ChemiDoc XRS+ System (Bio-Rad) and hemolytic zone areas measured using ImageJ. Images are representative of at least three independent repeats.

## α-toxin ELISA

The concentration of α-toxin in culture supernatants or lavage and organ samples was measured using an α-toxin-specific sandwich ELISA exactly as described previously[12,28]. Additional details are found in the Supplementary Methods. Experiments were repeated in triplicate (in vitro) or duplicate (in vivo) and data were combined and expressed as the mean + SEM.

## CFU enumeration

Microbial burdens of lavage fluid, homogenized kidneys and spleens of infected mice, or culture media were determined as previously described. Briefly, serial dilutions were plated onto YPD with 50 μg/mL chloramphenicol (for *C. albicans* enumeration) and TSA with 2.5 μg/mL amphotericin B (for *S. aureus* enumeration) via the drop-plate method. Plates were incubated overnight at 37 °C. Microbial burden was enumerated and expressed as CFU/mL. CFU values are representative of at least 3 independent repeats and are represented as mean ± SEM or median.

## Transcription factor mutant screen

*C. albicans* transcription factor mutants were screened for their ability to enhance *S. aureus agr* activity using the GFP-*agr* reporter assay protocol described previously. As the library control and mutant strains are arginine auxotrophs, 40 μg/mL arginine was supplemented in the culture medium. Fold-change fluorescence was calculated by comparing each library mutant to the monomicrobial *S. aureus* control. Hits were identified by using a 2-fold standard deviation cutoff as established by z-score analysis. Strains meeting these criteria were independently confirmed and used for further analyses.

## Ribose measurement by LC-MS/MS

For in vitro experiments, 495 μL of filtered culture supernatants, prepared ribose (Sigma, Cat. No. 1603108) standards (0.005–50 mg/mL), or water were mixed with 5 μL shikimic acid (Sigma, Cat No. 69686) solution as an internal standard (IS) prior to measurement. For in vivo samples, 495 μL of each prepared sample was directly used for the LC-MS/MS measurement after adding 5 μL IS solution. The standard concentration range for in vivo samples was from 0.0001 to 1 mg/mL. LC-MS/MS results were acquired using a Sciex (Framingham, MA) 5500 TripleQuad Mass Spectrometer coupled with Shimadzu (Columbia, MD) LC20ADXR binary pumps, Shimadzu SIL20ACXR autosampler and Shimadzu CTO20AC column oven. LC-MS/MS data was acquired using Analyst software (version 1.6.3) and analyzed using MultiQuant

software (version 3.0.2). To build the calibration curve, non-linear fittings were used for both in vitro and in vivo samples. All analyses were performed by the UTHSC Analytical Facility as fee-for-service. Data are presented as the mean + SEM.

## RNA isolation for transcriptional profiling
Monomicrobial cultures of *C. albicans* TF WT and Δ/Δ*zcf13* were grown as in the reporter assays in biological triplicate. At 16 h cultures were centrifuged and the cell pellets frozen at −80 °C. RNA was extracted following the hot acid phenol RNA isolation protocol and purified using the RNeasy Mini Kit (Qiagen, Cat No. 74104)[88]. RNA samples were treated with DNase I (Qiagen, Cat No. 79254) using the on-column digestion protocol prior to sequencing. *S. aureus* monomicrobial cultures were grown for 16 h in quadruplicate in the presence (0.5 mg/mL) or absence of exogenous ribose. Cells were centrifuged and pellets frozen at −80 °C. After thawing, samples were resuspended in 100 µl TE buffer pH 8 and RNA extraction performed according to the RNeasy Mini Kit (Qiagen, Cat No. 74104) protocol with few modifications. Cells were transferred to a Lysing Matrix B tube and bead beat with 0.1 mm beads for 60 s, followed by the addition of 650 µl of buffer RLT containing β-mercaptoethanol, and another 60 s of bead beating. The rest of the steps were followed according to manufacturer's instructions[89].

## RNA-sequencing and analysis
All library preparation, sequencing, and analyses were performed by Novogene as fee-for-service. Details regarding library preparation and raw data processing are found in the Supplementary Methods. Sequencing data were submitted to the Sequence Read Archive (BioProject identifiers PRJNA1074308 and PRJNA1074315). Differential expression analysis with biological replicates ($n = 3$–4) was performed using the DESeq2 and edgeR packages and analyzed using Benjamini-Hochberg's procedure to control for False Discovery Rate[90,91]. Genes showing 1.5-fold differential expression and adjusted $p$-value < 0.05 were considered significantly different. Heatmaps of differential expression data were constructed using Heatmapper[92]. Volcano plots were constructed using the MaGIC Volcano Plot Tool (https://volcano.bioinformagic.tools).

## Reverse transcription-quantitative PCR (qRT-PCR)
*C. albicans* was grown as described in the GFP reporter assay protocol above. RNA extraction was performed using the hot-acid phenol method described previously[93]. Removal of genomic DNA from the extracted RNA and first-strand cDNA synthesis was performed using TurboDNase (Invitrogen, Cat. No. AM22238) and the RevertAid RT Kit (Thermo Scientific, Cat. No. K1691) according to the manufacturer's protocol. cDNA (100 ng) was amplified with 2X Maxima SYBR Green/ROX qPCR Master Mix (Thermo Scientific, Cat. No. K0251) and gene-specific or Ca*ACT1* (housekeeping gene) primers found in Supplementary Table 3. Amplification was performed according to manufacturer's instructions using a Bio-Rad CFX96 Real-Time System. Expression levels of genes of interest were analyzed using the $2^{-\Delta\Delta CT}$ method to compare to WT strains and normalized to *ACT1*[94]. Data are depicted as mean + SEM.

## Statistical analyses and reproducibility
Experimental data was assessed for normality using a Shapiro-Wilk test. If normally distributed, unpaired two-tailed Student's t-test, multiple t-test, or a one-way ANOVA and Dunnett's post-tests were used. Nonparametric data was assessed using a Mann–Whitney test. Group numbers for animal studies were determined using a power analysis (α of 0.05, power of 80%). A Gehan-Breslow-Wilcoxon test was used to determine the significance of mortality. Specific tests are indicated in each figure legend. All statistical analyses were performed using GraphPad Prism v10.1.0. No data were excluded from analyses,

the experiments were not randomized, and investigators were not blinded during experiments or outcomes assessment.

## Figure construction
All graphs were constructed in GraphPad Prism v10.1.0, LivingImage, or Heatmapper. Biorender was used to construct Fig. 10g under a Creative Commons Attribution-NonCommercial-NoDerivs 4.0 International license (https://creativecommons.org/licenses/by-nc-nd/4.0/deed.en). Any adjustments to brightness or contrast were applied evenly across the entire image. High-resolution figures were rendered for publication using GraphPad Prism v10.1.0.

## Reporting summary
Further information on research design is available in the Nature Portfolio Reporting Summary linked to this article.

## Data availability
Source data are provided within this paper, in the Supplementary Information, or housed on the Sequence Read Archive (PRJNA1074308 and PRJNA1074315). Unique biological reagents created in this work can be obtained by contacting the corresponding author. Source data are provided with this paper.

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

## Acknowledgements

This work was supported by the National Institute of Allergy and Infectious Diseases (grants R01AI134796 to B.M.P.; R01AI116025 to M.C.N. and B.M.P.; R01AI145096 to P.L.F.; R01AI145992, R01AI161022,

and R01AI173795 to J.E.C.) and the UTHSC Center for Pediatric Experimental Therapeutics (to O.A.T.). The authors thank Dr. Dejian Ma (UTHSC Analytical Facility) for generous assistance with LC-MS studies. The LC-MS instrument was supported by NIH S10 grant 1S10OD016226-01A1. We also thank Ms. Jennifer Stabenow (UTHSC Regional Biocontainment Laboratory) for clinical chemistry expertise.

## Author contributions

B.M.P., S.P., O.A.T., and M.C.N. conceptualized the study. S.P. and O.A.T. collected the data. S.P., O.A.T., and B.M.P. analyzed the data. S.P., O.A.T., and B.M.P. prepared the manuscript. S.P., O.A.T., K.R.E., C.T., B.R.S., M.C.N., J.E.C., P.L.F., and B.M.P. contributed to manuscript review, editing, and gave final approval.

## Competing interests

The authors declare no competing interests.

## Additional information

[1]Department of Clinical Pharmacy and Translational Science, University of Tennessee Health Science Center, Memphis, TN, USA. [2]Integrated Program in Biomedical Sciences, University of Tennessee Health Science Center, Memphis, TN, USA. [3]Department of Pediatrics, Division of Pediatric Infectious Diseases, Vanderbilt University Medical Center, Nashville, TN, USA. [4]Early Vaccines and Immune Therapies, AstraZeneca, Gaithersburg, MD, USA. [5]Department of Microbiology and Immunology, School of Medicine, Tulane University, New Orleans, LA, USA. [6]Department of Pathology, Microbiology, and Immunology, Vanderbilt University Medical Center, Nashville, TN, USA. [7]Department of Biomedical Engineering, Vanderbilt University, Nashville, TN, USA. [8]Vanderbilt Institute for Infection, Immunology, and Inflammation (VI4), Vanderbilt University Medical Center, Nashville, TN, USA. [9]Department of Oral and Craniofacial Biology, Louisiana State University Health - School of Dentistry, New Orleans, LA, USA. [10]Department of Microbiology, Immunology, and Biochemistry, University of Tennessee Health Science Center, Memphis, TN, USA. [11]These authors contributed equally: Saikat Paul, Olivia A. Todd. ✉e-mail: brian.peters@uthsc.edu

