## [Peer Review File · Nature Communications]

REVIEWER COMMENTS

Reviewer #1 (Remarks to the Author):

The manuscript of Paul et al. describes a very nice genetic study of the mechanism underlying the synergistic interaction between *Candida albicans* and *Staphylococcus aureus* during intra-abdominal infection. They establish that expression of the *S aureus* toxin alpha toxin mediates this interaction. Using an impressive genetic screen and follow up studies they identify the relatively uncharacterized TF znf13 as required for the induction of alpha toxin by *C. albicans*. They then identify key targets of Zcf13 that mediate this activation.

The experiments were all well designed with extensive controls; the data is really spectacular and supports the conclusions. This is study is in the top percentile of candida pathogenesis manuscripts that I have read.

It will be of interest to a broad range of scientists involved in pathogenesis and shows the power of genetic analysis that can be brought to bear on questions of related to *Candida* pathogenesis.

I have only two minor suggestions.

1. The literature cited to support the clinical relevance of the *S aureus* *C. albicans* interaction focuses mainly on generic studies of polymicrobial infections. Some of the references do not directly relate to the content of the sentences they are associated with. For example, in lines 75-78 one of the references is to pediatric osteoarticular infections which has little to do with the study. I also could not find the data for increased mortality in ref 3 and 4. Since *S aureus* and *Candida albicans* are both common causes of peritoneal dialysis catheters it might be worth exploring that literature. A bit more attention to the references here is warranted since this is not an extensively studied process.

2. The indicators of statistical significance for the survival curves do not describe what groups are different. In some it is rather obvious that is relative to WT but for experiments with multiple experimental groups on the same graph (such as 10f) it is less clear.

Reviewer #2 (Remarks to the Author):

In a previous publication (Todd et al 2019), the authors described the synergistic lethality and organ damage caused by *S. aureus* in co-culture with *Candida albicans* in a murine intra-abdominal infection as a consequence of staphylococcal alpha-toxin (Hla) upregulation. Here the authors describe screening a bank of *C. albicans* transcription factor deletion mutants for the candidal factor(s) responsible for inducing hla gene expression. A *C. albicans* zcf13 mutant unable to enhance alpha-toxin production or lethal synergy was identified which in turn was shown to regulate genes involved in pentose sugar metabolism and in particular a ribokinase (RBK1) and a ribose transporter (HGT7). Overexpression of both RBK1 and HGT7 in a *C. albicans* zcf13 mutant restored enhanced agr expression, alpha-toxin production and synergistic lethality in co-culture with *S. aureus*. Since ribose inhibited agr and hence hla expression and as ribose is the second most abundant carbohydrate in human serum the authors concluded that they have uncovered a mechanism by which metabolic changes in a fungal commensal have a major impact on the virulence of a bacterial pathogen with serious consequences for the infected human host. This is an interesting story with generally sound experiments backing up the authors conclusions. However, the authors' experiments do not explain the mechanism by which co-culture of *S. aureus* with wild type *C. albicans* results in increased alpha-toxin production or how ribose down-regulates alpha-toxin production or how ribose inhibits staphylococcal growth. The authors speculate that it might involve staphylococcal regulators RpiRC or PurR given their known agr/exotoxin phenotypes. Can *S. aureus* reduce local ribose levels and so increase hla expression by inducing ribose uptake/metabolism in *C. albicans*?

Therefore, as it stands, although the authors describe some novel findings, they do not answer the questions that stimulated their original question although it could be argued that in the host, localized ribose levels may be reduced by *Candida* sufficiently to lead to the up-regulation of agr and hence hla in *S. aureus*.

Minor points

- 1.The *S. aureus* JE2 hla mutant used is a transposon insertion mutant not a deletion mutant and so should not be referred to as a deltahla mutant. Genetic complementation of this mutant in the experiments described is missing throughout.
- 2.Fig. 1 could be improved by including data for the *S. aureus* WT alone.
- 3.Growth curves - Y axes in Fig. 8c and Fig. S3b should be log scale.
- 4.Fig. 8b and 8c. Since agr induction is population density and growth rate dependent, is it possible that agr inhibition by ribose is due to an effect on growth rather than a specific inhibitory mechanism given that there is little difference between the concentrations of ribose required for agr and growth inhibition (0.5 and 1 mg/ml). For 8b, AFU should be divided by OD.
- 5.Fig. 8e is not a heatmap as stated in the legend but a volcano plot.

Reviewer #3 (Remarks to the Author):

Fungal bacterial polymicrobial infections has become an alarming healthcare challenge due to their ever rising incidence, increased mortalities and morbidities resulting significant patient discomfort, long hospital stays and economic losses. *Candida albicans* and *Staphylococcus aureus* are well known fungal and bacterial pathogens associated with such healthcare associated polymicrobial infections and their pathogenic synergies has troubled the healthcare team in managing the infections. In line with the growing interest in managing *C. albicans* – *S. aureus* polymicrobial infections, the study presented here investigated the molecular mechanism(s) by which the fungus enhances the lethality of the bacterial pathogen using a murine model of polymicrobial intra-abdominal infection (IAI). The study is comprehensive, well conducted and the data acquired are compelling. Manuscript is well-written and easy to follow. For further refining the manuscript, I have following comments for authors to address.

Introduction is clear, research question is defined and the importance of the question is highlighted.

Results:

α -toxin is responsible for exacerbated organ damage during polymicrobial IAI: Did the authors compare the increase in the liver and kidney biomarkers in polymicrobial infection with monomicrobial (*S. aureus* only) IAI to verify whether the impact is stronger in polymicrobial infection?

Line 135: 'despite similar α -toxin production from both strains (Fig. 2b)': Did authors confirm this statistically by comparing the alpha toxin concentrations between the two strains?

Line 161: 'Follow up assays....production (Fig 3c)': It is not clear in the figure whether the mutants were compared to TF WT+ SA or SA alone. Please clarify. Also, please state the rationale behind colour coding in Fig 3b-3e.

Authors have used different time points e.g. 8h in Figure 2, 16h in Figure 3 and 8h in figure for as to asses microbial burden, toxin production etc. As per Figure 1, it appears that the significant impact on toxin production of *S. aureus* was observed at/after 12h of co-infection. I am curious as to whether the investigators may have missed some important observations by choosing earlier time points than 12h when assessing various parameters as described above.

Line 175-a77: 'Despite... strain (fig 4c)': As per Fig 4c, the difference of alpha toxin production between SA with zcf13Δ/Δ mutant and with TF WT is insignificant. I believe this statement regarding spleen is inaccurate based on that.

Lines 177-178: Since sfl1Δ/Δ elicited 50% mortality, did authors consider evaluating its impact similar to what has been done with zcf13Δ/Δ ?

Lines 184- 185: 'No major.... Co-culture (fig 5c)': There is a significant reduction in the CFU of the revertant strain when co-cultured with SA. Did authors take this in to consideration when making comparisons of AFU and alpha toxin concentrations?

Figures are well prepared and presented. However, it is important to state what experimental group/condition(s) has/have been used as the control when making statistical comparisons. All the figures presented with statistical significance have this information missing. Please carefully go through all figures and include this information either in the figure caption or in the main text.

Please include a better quality image for Figure 7a.

Discussion:

Lines 379-382: 'interestingly... cerevisiae.': While presented data in this study suggest that the ribose catabolism in *C. albicans* has a significant role in increased lethality of *S. aureus* - *C. albicans* coinfections by enhancing alpha toxin production, ribose would unlikely be the only carbon source or the primary carbon source available in the site of infection of IAI. How would the availability of other carbon sources impact on the suggested mechanism of increased lethality in this study? Are there any other mechanistic interplays of *C. albicans* -*S. aureus* polymicrobial interactions that need to be considered?

Methodology is clearly outlined and the supplementary information are appropriate.

Point-by-point responses to reviewer comments

Reviewer #1 (Remarks to the Author):

The manuscript of Paul et al. describes a very nice genetic study of the mechanism underlying the synergistic interaction between *Candida albicans* and *Staphylococcus aureus* during intra-abdominal infection. They establish that expression of the *S aureus* toxin alpha toxin mediates this interaction. Using an impressive genetic screen and follow up studies they identify the relatively uncharacterized TF znf13 as required for the induction of alpha toxin by *C. albicans*. They then identify key targets of Zcf13 that mediate this activation.

The experiments were all well designed with extensive controls; the data is really spectacular and supports the conclusions. This study is in the top percentile of candida pathogenesis manuscripts that I have read.

It will be of interest to a broad range of scientists involved in pathogenesis and shows the power of genetic analysis that can be brought to bear on questions of related to *Candida* pathogenesis.

We thank the reviewer for their supportive comments of our manuscript.

I have only two minor suggestions.

1. The literature cited to support the clinical relevance of the *S aureus* *C. albicans* interaction focuses mainly on generic studies of polymicrobial infections. Some of the references do not directly relate to the content of the sentences they are associated with. For example, in lines 75-78 one of the references is to pediatric osteoarticular infections which has little to do with the study. I also could not find the data for increased mortality in ref 3 and 4. Since *S aureus* and *Candida albicans* are both common causes of peritoneal dialysis catheters it might be worth exploring that literature. A bit more attention to the references here is warranted since this is not an extensively studied process.

We completely agree with the reviewer and apologize for not including more appropriate references in the initial version. We have now replaced with references indicating worsened outcomes during polymicrobial as compared to monomicrobial infection and high incidence of staphylococci and fungi during peritoneal dialysis-related and disseminated infection. The updates are found on lines 78-81.

2. The indicators of statistical significance for the survival curves do not describe what groups are different. In some it is rather obvious that is relative to WT but for experiments with multiple experimental groups on the same graph (such as 10f) it is less clear.

Thank you for this important comment. We have now explicitly described comparators used for statistical testing in each figure legend.

Reviewer #2 (Remarks to the Author):

In a previous publication (Todd et al 2019), the authors described the synergistic lethality and organ damage caused by *S. aureus* in co-culture with *Candida albicans* in a murine intra-abdominal infection as a consequence of staphylococcal alpha-toxin (Hla) upregulation. Here the authors describe screening a bank of *C. albicans* transcription factor deletion mutants for the 2andida factor(s) responsible for inducing hla gene expression. A *C. albicans* zcf13 mutant unable to enhance alpha-toxin production or lethal synergy was identified which in turn was shown to regulate genes involved in pentose sugar metabolism and in particular a ribokinase (RBK1) and a ribose transporter (HGT7). Overexpression of both RBK1 and HGT7 in a *C. albicans* zcf13 mutant restored enhanced agr expression, alpha-toxin production and synergistic lethality in co-culture with *S. aureus*. Since ribose inhibited agr and hence hla expression and as ribose is the second most abundant carbohydrate in human serum the authors concluded that they have uncovered a mechanism by which metabolic changes in a fungal commensal have a major impact on the virulence of a bacterial pathogen with serious consequences for the infected human host. This is an interesting story with generally sound experiments backing up the authors conclusions. However, the authors' experiments do not explain the mechanism by which co-culture of *S. aureus* with wild type *C. albicans* results in increased alpha-toxin production or how ribose down-regulates alpha-toxin production or how ribose inhibits staphylococcal growth. The authors speculate that it might involve staphylococcal regulators RpiRC or PurR given their known agr/exotoxin phenotypes. Can *S. aureus* reduce local ribose levels and so increase hla expression by inducing ribose uptake/metabolism in *C. albicans*?

We appreciate the reviewer's comments regarding ribose-mediated *agr* repression during in vitro co-culture. To interrogate this further, we measured ribose levels in TSBg media and peritoneal lavage fluid obtained from naïve (i.e., uninfected) mice. We serendipitously observed that ribose levels in TSBg mirror that of recovered lavage fluid. Thus, in vitro conditions mimic the ribose environment in vivo. Moreover, ribose levels found in the lavage fluid of *S. aureus* mono-infected are not significantly different than those from naïve mice, further suggesting that *C. albicans* (and not *S. aureus*) is driving ribose depletion via the mechanisms supported by the underlying data. We have included this data as new Supplementary Fig. 5 and provided a description at lines 263 and 270 in the text.

Therefore, as it stands, although the authors describe some novel findings, they do not answer the questions that stimulated their original question although it could be argued that in the host, localized ribose levels may be reduced by Candida sufficiently to lead to the up-regulation of agr and hence hla in S. aureus.

We thank the reviewer for their comment. It is difficult to precisely measure microenvironmental levels of ribose in vivo, but as the reviewer posits, we also suspect that local depletion of ribose levels in the peritoneal cavity may more strongly activate staphylococcal *agr* in a site-specific manner.

Minor points

1. The *S. aureus* JE2 hla mutant used is a transposon insertion mutant not a deletion mutant and so should not be referred to as a deltahla mutant. Genetic complementation of this mutant in the experiments described is missing throughout.

We have updated Supplementary Table 1 and Figures 1 and 2 to indicate that this is indeed the *hla::bursa* strain. We have complemented this mutant in Figure 2 and show that it restores in

vitro hemolysis and synergistic lethality in vivo (Figure 2). These results are similar to its complementation previously reported in PMID: 31164467.

2.Fig. 1 could be improved by including data for the S. aureus WT alone.

We thank the reviewer for this point. While we agree that the data could be stronger with the S. aureus mono-infection, the main point of the figure is to show dependence on *hla* (even if partial) during co-infection. As we had generated this data several years ago, and our institution has since updated equipment, we would need to repeat the entire experiment again to ensure scalable data across groups. This would entail the possibly unjustified use of 72 mice (3 groups x 3 time points x 8 mice/group) for what we feel would not impact interpretation or overall conclusion of our manuscript.

3.Growth curves - Y axes in Fig. 8c and Fig. S3b should be log scale.

We have moved old 8c to Supplementary Fig. 3b and prior Supplementary Fig. 3b to 4b. Both graphs have been plotted on a log scale as suggested.

4.Fig. 8b and 8c. Since agr induction is population density and growth rate dependent, is it possible that agr inhibition by ribose is due to an effect on growth rather than a specific inhibitory mechanism given that there is little difference between the concentrations of ribose required for agr and growth inhibition (0.5 and 1 mg/ml). For 8b, AFU should be divided by OD.

We thank the reviewer for their helpful suggestion here. Along with including a few additional ribose doses (0.2, 0.35 mg/mL), we also calculated normalized fluorescence by dividing the OD600 readings. If fluorescence was merely density-dependent, the lines should generally overlap. However, we see clear separation of normalized fluorescence signal that is dose-dependent with respect to ribose. We have now removed prior Figures 8b and 8c and replaced with a new Figure 8b showing normalized fluorescence. Subsequent panels have been renumbered accordingly. Updates are found on lines 233-236.

5.Fig. 8e is not a heatmap as stated in the legend but a volcano plot.

We thank the reviewer for catching this error. We originally had formatted the data as a heatmap but switched to a volcano plot in later revisions. We have now corrected appropriately.

Reviewer #3 (Remarks to the Author):

Fungal bacterial polymicrobial infections has become an alarming healthcare challenge due to their ever rising incidence, increased mortalities and morbidities resulting significant patient discomfort, long hospital stays and economic losses. Candida albicans and Staphylococcus aureus are well known fungal and bacterial pathogens associated with such healthcare associated polymicrobial infections and their pathogenic synergies has troubled the healthcare team in managing the infections. In line with the growing interest in managing C. albicans – S. aureus polymicrobial infections, the study presented here investigated the molecular mechanism(s) by which the fungus enhances the lethality of the bacterial pathogen using a murine model of polymicrobial intra-abdominal infection (IAI). The study is comprehensive, well conducted and the data acquired are compelling. Manuscript is well-written and easy to

follow. For further refining the manuscript, I have following comments for authors to address.

Introduction is clear, research question is defined and the importance of the question is highlighted.

We thank the reviewer for their supportive comments of our work.

Results:

α -toxin is responsible for exacerbated organ damage during polymicrobial IAI: Did the authors compare the increase in the liver and kidney biomarkers in polymicrobial infection with monomicrobial (*S. aureus* only) IAI to verify whether the impact is stronger in polymicrobial infection?

We thank the reviewer for this point. While we agree that the data could be stronger with the *S. aureus* mono-infection, the main point of the figure is to show dependence on *hla* (even if partial) during co-infection. As we had generated this data several years ago, and our institution has since updated equipment, we would need to repeat the entire experiment over again to ensure scalable data across groups. This would entail the possibly unjustified use of 72 mice (3 groups x 3 time points x 8 mice/group) for what we feel would not impact interpretation or overall conclusion of our manuscript.

Line 135: 'despite similar α -toxin production from both strains (Fig. 2b)': Did authors confirm this statistically by comparing the alpha toxin concentrations between the two strains?

We have now conducted a two-tailed Student's t-test to show that these are not significantly different and Figure 2b has been updated with new statistical markers accordingly.

Line 161: 'Follow up assays....production (Fig 3c)': It is not clear in the figure whether the mutants were compared to TF WT+ SA or SA alone. Please clarify. Also, please state the rationale behind colour coding in Fig 3b-3e.

In this case, all strains were compared to SA alone. However, we have now provided descriptors in all figure legends to explicitly state the comparators used. We have attempted to keep the color schema similar throughout the manuscript where SA is yellow, CA WT+SA is green, and *zcf13* Δ/Δ +SA is teal. Although the color coding is somewhat arbitrary initially in Figure 3, hopefully it becomes clearer to the reader as they move through the manuscript.

Authors have used different time points e.g. 8h in Figure 2, 16h in Figure 3 and 8h in figure for as to asses microbial burden, toxin production etc. As per Figure 1, it appears that the significant impact on toxin production of *S. aureus* was observed at/after 12h of co-infection. I am curious as to whether the investigators may have missed some important observations by choosing earlier time points than 12h when assessing various parameters as described above.

We thank the reviewer for their thoughtful comment. It is possible that we do miss relevant infectious processes by looking early (8 h post-infection). However, the reason for selecting the 8 h time point is because lethality is variable starting beyond 10 h p.i. Often, some mice from the same cohort succumb or are significantly moribund by 10-12 h p.i., while the others noticeably less so. Therefore, we wanted to capture a time point where initial uniform morbidity was

observed to allow for the recovery of appropriately powered experimental data. However, we agree that it would be interesting to begin investigating dysregulated biological processes nearer the endpoint.

Line 175-a77: 'Despite... strain (fig 4c)': As per Fig 4c, the difference of alpha toxin production between SA with *zcf13Δ/Δ* mutant and with TF WT is insignificant. I believe this statement regarding spleen is inaccurate based on that.

We apologize for this oversight and have updated the text in line 181 to clarify. Although the data was nearly significant, we have removed "spleen" for accuracy.

Lines 177-178: Since *sfl1Δ/Δ* elicited 50% mortality, did authors consider evaluating its impact similar to what has been done with *zcf13Δ/Δ* ?

We appreciate the reviewer's comment. We have not extensively explored the *sfl1Δ/Δ* as we were focused on *zcf13Δ/Δ* that gave a more striking phenotype. It would be interesting to begin characterizing this mutant as well (beyond its known role as a morphogenetic regulator) but we feel it is beyond the scope of this current study.

Lines 184- 185: 'No major.... Co-culture (fig 5c)': There is a significant reduction in the CFU of the revertant strain when co-cultured with SA. Did authors take this into consideration when making comparisons of AFU and alpha toxin concentrations?

We thank the reviewer for their thoughtful comment. We did observe a slight decrease in CFU of the *ZCF13* revertant in 2 of the 3 in vitro experiments that was statistically significant. We are not sure of the biological relevance of this result as we did not observe any difference in fungal burden in vivo compared to the WT or mutant strain.

Figures are well prepared and presented. However, it is important to state what experimental group/condition(s) has/have been used as the control when making statistical comparisons. All the figures presented with statistical significance have this information missing. Please carefully go through all figures and include this information either in the figure caption or in the main text.

We thank the reviewer for this important comment. We have updated all figure legends to include a description of the comparator used for statistical analysis.

Please include a better quality image for Figure 7a.

We have included a more resolved image of the heatmap presented in Figure 7a.

Discussion:

Lines 379-382: 'interestingly... cerevisiae.': While presented data in this study suggest that the ribose catabolism in *C. albicans* has a significant role in increased lethality of *S. aureus* - *C. albicans* coinfections by enhancing alpha toxin production, ribose would unlikely be the only carbon source or the primary carbon source available in the site of infection of IAI. How would the availability of other carbon sources impact on the suggested mechanism of increased lethality in this study? Are there any other mechanistic interplays of *C. albicans* -*S. aureus* polymicrobial interactions that need to be considered?

This is an important point. While our work suggests that ribose is an important player in modulating *agr* activity in vitro and in vivo, it is equally possible that other carbon sources in vivo could exert a similar effect. We have added a statement to address this possibility in lines 393-397.

Methodology is clearly outlined and the supplementary information are appropriate.

REVIEWERS' COMMENTS

Reviewer #2 (Remarks to the Author):

The authors have addressed all my concerns and I am very happy with the revised manuscript.

Reviewer #3 (Remarks to the Author):

The authors have addressed my comments satisfactorily.